# Evolution of gene expression in seasonal environments

**Shuichi N Kudo[1], Yuka Ikezaki[2], Junko Kusumi[3], Hideki Hirakawa[4,5], Sachiko Isobe[4,6], Akiko Satake[2]***

[1]Graduate School of Systems Life Sciences, Kyushu University, Fukuoka, Japan; [2]Department of Biology, Faculty of Science, Kyushu University, Fukuoka, Japan; [3]Department of Environmental Changes, Faculty of Social and Cultural Studies, Kyushu University, Fukuoka, Japan; [4]Kazusa DNA Research Institute, Chiba, Japan; [5]Graduate School of Bioresource and Bioenvironmental Sciences, Kyushu University, Fukuoka, Japan; [6]Graduate School of Agricultural and Life Sciences, The University of Tokyo, Tokyo, Japan

*For correspondence:
satake.akiko.269@m.kyushu-u.ac.jp

## eLife Assessment

The authors collected time-course RNA-seq data from four tree species in natural environments and analyzed seasonal patterns of gene expression. This **fundamental** study substantially advances our understanding of how seasonal environments shape gene expression. The evolutionary effects of seasonal environments on gene expression are rarely studied at this scale and the dataset is extensive. The evidence supporting the conclusions is **compelling**, with caveats and limitations clearly described. The work will be of broad interest to colleagues studying evolution and gene expression.

**Abstract** The biological activities of organisms are closely linked to seasonality. Phenology, the temporal orchestration of biological activities, is governed by gene expression, yet the evolutionary dynamics underlying seasonal gene expression remain unclear. To investigate these dynamics, we compared genome-wide expression dynamics (molecular phenology) in four dominant evergreen Fagaceae species in Asia (*Quercus glauca*, *Q. acuta*, *Lithocarpus edulis*, and *L. glaber*), using leaf and bud tissues over two seasonal cycles. We assembled high-quality reference genomes, identifying 11749 single-copy orthologous genes. Seasonal transcriptomic profiling of these orthologous genes revealed highly conserved gene expression across species in winter when temperatures fall below ~10 °C. Rhythmic gene expression with significant periodic oscillations was more prevalent in buds (51.9%) than in leaves (40.6%), with most rhythmic genes (78.4–92.0%) exhibiting annual periodicity, while a smaller fraction (1.2–11.9%) followed half-annual cycles. The seasonal peaks of rhythmic genes were highly synchronized across species in winter but diverged during the growing season, reflecting species-specific timing of leaf flushing and flowering. These findings suggest that the four species share a common molecular calendar in winter, which constrains the evolution of gene expression under seasonal environments.

## Introduction

Fundamental biological processes, such as growth, mortality, and reproduction, rarely remain constant over time; instead, they exhibit pronounced variation in response to seasonal environmental changes (*Fretwell, 1972*). This temporal organization of a series of biological activities, known as phenology, plays a critical role in the evolution of life histories in response to seasonal fluctuating environments (*Forrest and Miller-Rushing, 2010*; *Morellato et al., 2013*; *Piao et al., 2019*; *Schwartz, 2013*).

Phenology is regulated by the coordinated expression of genes that enable organisms to exhibit essential functions at the appropriate times (*Kudoh, 2016*; *Satake et al., 2024a*; *Satake et al., 2022*). However, gene expression dynamics underpinning seasonal responses in organisms remain not fully understood.

To explore the evolution of gene expression, transcriptome analyses are widely employed to compare transcript profiles across organs, sexes, species, and developmental stages within a given environment. Such studies have been conducted in mammals (*Brawand et al., 2011*; *Cardoso-Moreira et al., 2019*; *Naqvi et al., 2019*), Drosophila (*Lemos et al., 2005*; *Nuzhdin et al., 2004*), and Tanzanian cichlids (*El Taher et al., 2021*), providing key insights into the evolutionary dynamics of gene expression and its central role in shaping phenotypic diversity. These studies have revealed that gene expression profiles tend to be more conserved across species within the same tissue, whereas the evolutionary rate of gene expression varies among organs. In mammals, for instance, nervous tissue exhibits a slower rate of gene expression change compared to the testis (*Brawand et al., 2011*; *Cardoso-Moreira et al., 2019*; *Naqvi et al., 2019*), suggesting the presence of tissue-specific functional constraints that influence the evolution of gene expression.

In addition to tissue-dependent constraints, seasonal environmental changes may play a crucial role in shaping the evolution of gene expression, particularly by influencing the temporal coordination of gene expression under natural conditions. Gene expression responses to certain seasons may be more resistant to change, whereas in other seasons, divergence in gene expression timing across species may occur, contributing to temporal niche differentiation. Despite its potential importance, the influence of seasonal environmental fluctuations on the gene expression evolution remains poorly understood, largely because most transcriptomic analyses have been conducted under tightly controlled laboratory conditions.

To investigate the evolution of gene expression in seasonal environments, we performed a molecular phenology analysis and phenological observation of leaf flushing and flowering in Fagaceae tree species, which are widely distributed across the Northern Hemisphere, with its center of diversity in subtropical Southeast Asia (*Hipp et al., 2020*; *Kremer et al., 2012*; *Manos et al., 2001*; *Zhou et al., 2022*). Molecular phenology refers to transcriptome analysis conducted under natural fluctuating conditions (*Komoto et al., 2024*; *Kudoh, 2016*; *Nagano et al., 2019*; *Satake et al., 2023*) and provides a valuable framework for studying gene–environment interactions and their role in orchestrating the temporal biological functions essential for adaptation to environmental fluctuations (*Satake et al., 2022*). In this study, we selected four evergreen Fagaceae species—two *Quercus* species (*Q. glauca* and *Q. acuta*) and two *Lithocarpus* species (*L. edulis* and *L. glaber*)—that co-occur in the same locality in Japan. *Q. glauca*, *L. edulis,* and *L. glaber* inhabit nearly identical environment, while *Q. acuta* is restricted to the high-altitude regions with slightly colder environment compared to the other three species. Leaf flushing and flowering phenology vary across species, with *Q. glauca* and *Q. acuta* flowering in early spring, almost simultaneously with leaf flushing, *L. edulis* flowering in late spring, and *L. glaber* flowering in autumn, approximately one to two months after leaf flushing. This setting enabled us to compare seasonal gene expression profiles across species with diverse phenology, both within and between genera, under nearly identical environmental fluctuations. By assessing and comparing seasonal transcriptomic dynamics across Fagaceae species, our study provides new insights into the evolution of gene expression and adaptation to seasonal natural environments.

## Results

### Genome assemblies of *Q. glauca* and *L. edulis*

To quantify transcript abundance of non-model Fagaceae species, we first constructed reference genomes for *Q. glauca*, as a representative of genus *Quercus*, and *L. edulis,* as a representative of genus *Lithocarpus* (*Figure 1A*). To obtain high-quality chromosome-level reference genomes, we conducted a hybrid genome assembly for the HiFi reads and the high-throughput chromosome conformation capture (Hi-C) reads (*Supplementary file 1*). The resultant scaffolds were 867.7 and 835.7 Mb and consisted of 797 and 750 scaffolds including 12 chromosome-level scaffolds with contig N50 of 71.7 Mb and 68.8 Mb for *Q. glauca* and *L. edulis*, respectively (*Supplementary file 1*). Approximately 94.8% and 97.2% of the final assemblies were assigned to 12 pseudochromosomes of *Q. glauca* and *L. edulis*, respectively (*Figure 1B*). The number of chromosome-scale scaffolds were

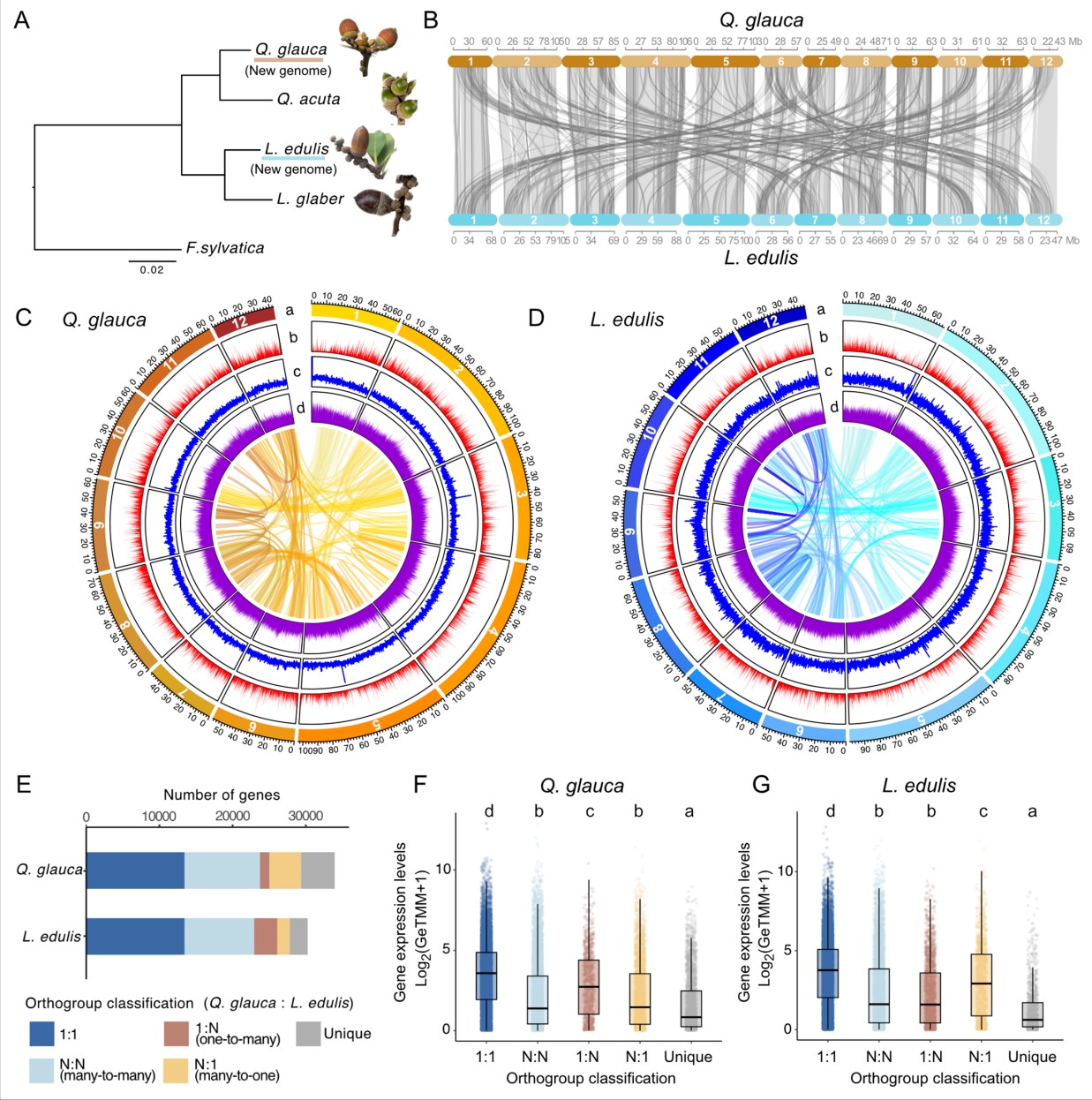

**Figure 1.** Synteny and distribution of genomic features of *Q. glauca* and *L. edulis* genomes. (**A**) A phylogenetic tree of five Fagaceae species from three genera, *Quercus*, *Lithocarpus*, and *Fagus*. Bootstrap values for all branches are 100. (**B**) Synteny relationship between the *Q. glauca* and *L. edulis* genomes. The collinear blocks within the genomes were displayed by lines. Black lines show reverse alignment and gray lines represent regular. Numbers correspond to the 12 chromosomes. (**C, D**) Circos plots of the *Q. glauca* (**C**) and *L. edulis* genomes (**D**). Lanes depict circular representation of chromosome length (Mbp) (a), gene density (b), GC density (c), and repeat (d). Lines in the inner circle represent links between synteny-selected paralogs. (**E**) Orthogroup classification based on the orthogroup size difference between the *Q. glauca* and *L. edulis* genomes. The bar plot illustrates the number of genes in gene families categorized into 1:1 orthologs, N:N (many-to-many), 1:N (one-to-many), N:1 (many-to-one), and species-specific unique genes. (**F, G**) Mean gene expression levels in each copy number variation category for *Q. glauca* (**F**) and *L. edulis* (**G**). Different letters denote statistically significant differences ($p<0.05$, Steel-Dwass test).

identical to the known numbers in other *Quercus* (**Zoldos et al., 1999**) and *Lithocarpus* species (**Liu et al., 2023**). The genomes were estimated to contain 39758 and 34059 protein-coding genes (33947 and 30246 genes were annotated within 12 chromosomes), covering 98.3% and 95.0% of complete Benchmarking Universal Single Copy Orthologs (BUSCO) genes (eudicots_odb10) for *Q. glauca* and *L. edulis*, respectively. The genome sizes of *Q. glauca* and *L. edulis* were estimated to be 891.2 Mb

and 868.7 Mb, respectively, using KmerGenie (*Chikhi and Medvedev, 2014*) and GenomeScope2 (*Ranallo-Benavidez et al., 2020*).

Overall, the abundance and distribution of protein-coding genes, GC content, and repeat sequences (*Figure 1C and D*; *Supplementary file 1*) and syntenic structure (*Figure 1B*) were similar between *Q. glauca* and *L. edulis*. This pattern of synteny conservation appears to be broadly shared across Fagaceae, except for the genus *Fagus*, which shows a distinct genome structure (*Ikezaki et al., 2025*). The homology between different chromosomes within the genome suggests that small-scale segmental duplications likely occurred, as previously proposed (*Chen et al., 2014*). Using OrthoFinder2 (*Emms and Kelly, 2019*), we clustered genes from the two newly assembled genomes into orthologous groups. We identified 13427 single-copy orthologs, enabling pairwise comparisons between species (*Figure 1E*).

## Seasonal gene expression dynamics

To characterize the evolution of seasonal gene expressions, we generated a total of 483 transcriptome profiles every four weeks over two years using leaf and bud tissues from *Q. glauca*, *Q. acuta*, *L. edulis*, and *L. glaber* during 2021–2023. Reads from the two *Quercus* species were mapped to the newly constructed *Q. glauca* reference genome, achieving a mean mapping rate of 92.6±2.2% and 88.6±2.3% for *Q. glauca* and *Q. acuta*, respectively. Similarly, reads from the two *Lithocarpus* species were mapped to the newly constructed *L. edulis* reference genome, with a mean mapping rate of 89.3±2.7% and 84.3±5.4% for *L. edulis* and *L. glaber*, respectively. The number of reads uniquely mapped to the reference genome was 19.9 million reads per sample, on average. We used single-copy orthologous genes for further analyses because these genes exhibited significantly higher average expression levels than multi-copy genes (Steel-Dwass test, $p<0.05$; *Figure 1F and G*). Among the single-copy orthologous genes, we excluded genes with low expression levels (mean of Reads Per Kilobase (RPK) across samples for each species <1) and normalized the transcript count data using the GeTMM method (*Smid et al., 2018*). This process resulted in a high-quality expression dataset for 11749 genes (*Figure 2—figure supplement 1A*). For further analyses, we used the mean expression levels across three biological replicates (three individuals per species), except for *L. edulis*, where only one individual was available. We confirmed that expression patterns of individual genes were highly consistent across all three biological replicates (*Figure 2—figure supplement 2*) in *Q. glauca* (Pearson's correlation $r=0.947$), *Q. acuta* ($r=0.948$), and *L. glaber* ($r=0.948$).

To gain an initial overview of gene expression dynamics in a seasonal environment, we assessed the relative magnitude of seasonal variation in expression levels for each gene by calculating the standard deviation divided by the mean for each gene ($\sigma_i/\mu_i$) and compared it to the overall variation across all genes in the genome ($\sigma_g/\mu_g$) (*Figure 2—figure supplement 1B*). A prior study using 3025 yeast RNA-seq experiments reported that the dynamic range of individual genes across experiments was considerably smaller than the overall genomic variation under laboratory conditions (*Zrimec et al., 2021*). Consistently, our results demonstrate that despite substantial seasonal fluctuations in environmental conditions (*Figure 2—figure supplement 3*), individual gene expression remains relatively conserved. On average, the magnitude of genome-wide variation was 15.8 times greater than the seasonal variation observed for individual genes (*Figure 2—figure supplement 1B*).

To assess the similarity of gene expression profiles across tissues, species, and seasons, we performed a hierarchical clustering of seasonal transcriptome profiles from leaf and bud tissues across four species and identified five distinct clusters (cluster L1, L2, B1–B3) according to the Elbow method (*Figure 2A*; *Figure 2—figure supplement 4A*; *Supplementary file 2*). Transcriptional profiles were first grouped into leaves (clusters L1 and L2) and buds (clusters B1–B3), irrespective of seasons and species (*Figure 2A*), except that newly flushed leaf samples were grouped closely with buds (*Figure 2A*; *Figure 2—figure supplement 5*). Within the leaf cluster, transcriptome profiles were first clustered by genus and then by species in *Lithocarpus* (cluster L1; *Figure 2A*). In contrast, samples from *Quercus* species formed a mixture of two species (cluster L2; *Figure 2A*). Notably, the winter samples of *Q. acuta*, collected during periods of low mean temperatures, were distinctly separated from those collected in other seasons (cluster L2; *Figure 2A*). Within the bud cluster, transcriptome profiles were predominantly grouped by season—winter (cluster B3; November to February, with the exception of a March sample from *L. edulis*; *Supplementary file 2*) versus the other growing seasons (cluster B1 and B2; March to November; *Supplementary file 2*)—and subsequently organized by

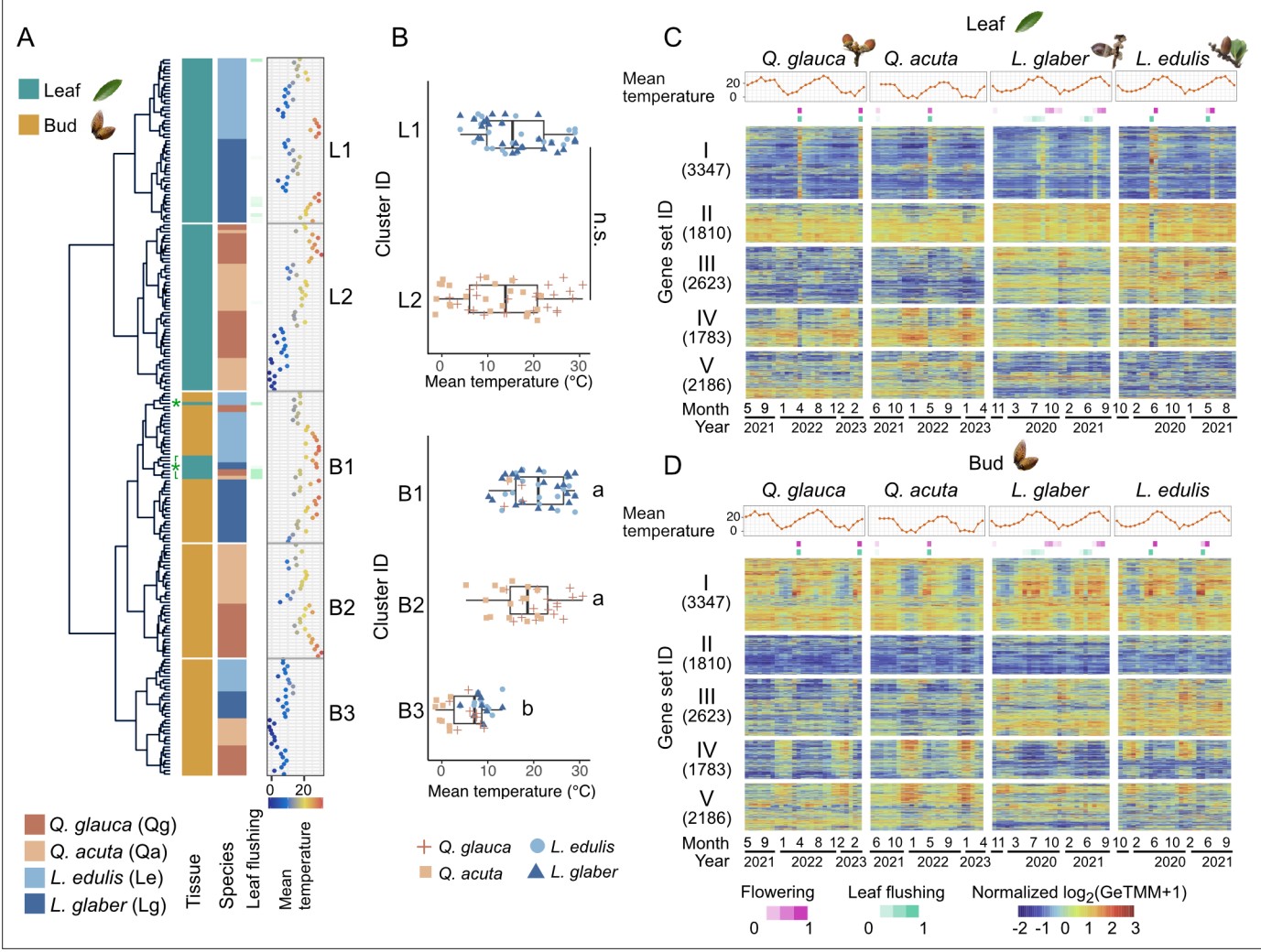

**Figure 2.** Seasonal gene expression dynamics. (**A**) Hierarchical clustering of 213 time points encompassing two tissues (leaf and bud) and four species (*Q. glauca*, *Q. acuta*, *L. edulis*, and *L. glaber*), collected over two years. Newly flushed leaf samples are indicated as asterisks. (**B**) Distribution of daily mean temperature on each sampling date across clusters, from L1 (top) to B3 (bottom). The Mann-Whitney U test was used to compare daily mean temperature between cluster L1 and L2 ($p$=0.136). The Steel-Dwass test was applied for multiple comparisons across clusters B1–B3 ($p<0.05$). Different letters denote statistically significant differences ($p<0.05$). (**C, D**) Hierarchical clustering of 11,749 genes based on the similarity of gene expression dynamics. Daily mean temperature, flowering, and leaf flushing phenology (proportion) are shown in the upper three panels of the heatmaps.

The online version of this article includes the following figure supplement(s) for figure 2:

**Figure supplement 1.** Dynamic range of gene expression in seasonal environments.

**Figure supplement 2.** Comparison of gene expression across individuals.

**Figure supplement 3.** Seasonal fluctuations in environmental factors at the three study sites.

**Figure supplement 4.** Determining the optimal number of clusters in hierarchical clustering.

**Figure supplement 5.** Phenological observations of leaf flushing, flowering (male and female flowers), and winter buds of the four target species.

**Figure supplement 6.** Seasonal gene expression dynamics observed within the same sampling period (from May 2021 to October 2022).

**Figure supplement 7.** Seasonal gene expression dynamics under the identical environmental condition.

**Figure supplement 8.** Distribution of environmental factors across clusters.

**Figure supplement 9.** 3D plot of PC1–PC3 resulting from the principal component analysis (PCA) of 11,749 orthologous genes in four Fagaceae species.

genus and species. This suggests that the winter season exerts a stronger influence on gene expression in buds than phylogenetic relationships. We confirmed the robustness of our findings to differences in sampling periods between *Quercus* and *Lithocarpus* species by reanalyzing expression data limited to common sampling periods (*Figure 2—figure supplement 6*). Even when using only the data from this shared period, the clustering patterns were consistently reproduced. To assess whether the higher-elevation site of *Q. acuta* introduced confounding environmental effects, we reanalyzed the data after excluding this species. Hierarchical clustering still revealed that winter bud samples formed a distinct cluster regardless of species identity (*Figure 2—figure supplement 7*), consistent with our original findings.

To assess the biological relevance of the clusters identified, we tested whether the temperatures and photoperiod at the time of sample collection differed significantly among the clusters. Within the bud cluster, transcriptional profiles in the winter cluster (cluster B3) were associated with significantly lower temperatures—approximately below 10°C—compared to those in other seasons (*Figure 2B*; *Figure 2—figure supplement 8A*). In contrast, no such temperature-related distinction was evident within the leaf cluster (*Figure 2B*). Additionally, photoperiods were markedly shorter in the winter cluster in buds (*Figure 2—figure supplement 8B*). These findings suggest that the buds of the four species exhibit a conserved transcriptional response to environmental cues, namely cold temperatures and short photoperiods, characteristic of the winter season. To further investigate the relationships among all samples, we conducted a principal component analysis (PCA). The first principal component (PC1) clearly separated samples by tissue type, while the second (PC2) and the third principal component (PC3) gradually differentiated winter samples from those collected during other seasons (*Figure 2—figure supplement 9*).

To investigate seasonal expression dynamics of each gene, we conducted hierarchical clustering on the transcriptional dynamics of 11,749 genes and identified five distinct gene sets characterizing different transcriptional modes (gene set I–V; *Figure 2C and D*; *Figure 2—figure supplement 4B*; *Supplementary file 3*). Gene set I exhibited predominant expression in buds, with seasonal peaks during summer, whereas gene set II was primarily expressed in leaves, highlighting clear tissue specificity (*Figure 2C and D*). Gene sets III and V demonstrated genus-specific expression patterns, with higher expression levels observed in *Lithocarpus* and *Quercus*, respectively (*Figure 2C and D*). In contrast, gene set IV was expressed during winter across both leaf and bud tissues (*Figure 2C and D*).

## Detection of rhythmic genes

To identify genes revealing seasonal expression dynamics and determine the period and seasonal peaks in gene expression, we applied the RAIN algorithm (*Thaben and Westermark, 2014*). This approach enabled us to detect genes with rhythmic expression (hereafter rhythmic genes) and analyze their periods and phases (peak months). Our analysis revealed that 44.6–58.8% of genes in bud tissues and 33.9–45.9% in leaf tissues exhibited significant rhythmic expression across species (*Figure 3—figure supplement 1A*; *Supplementary file 4*). Notably, bud tissues demonstrated a higher proportion of rhythmic genes, with 51.9% across four species on average, compared to 40.6% in leaf tissues (*Figure 3—figure supplement 1A*).

Among the rhythmic genes, the majority (78.4–92.0%) exhibited annual periodicity, with a period of 12±1 months observed across species and tissues (*Figure 3A and B*, *Supplementary file 4*). Additionally, hundreds of genes (1.2–11.9%) displayed half-annual periodicity (6±1 month) with these genes being most frequently observed in the bud samples of *L. edulis* (*Figure 3A and B*). The fraction of genes exhibiting other periodicities was relatively small (5.7–14.1%; *Figure 3A and B*). Based on the observed period distributions, we classified genes with periods within the range of 8–16 months as exhibiting annual periodicity (96.0%) and those with periods shorter than 8 months as having half-annual periodicity (4.0%), accounting for detection noise (*Figure 3—figure supplement 1A*). Rhythmic genes with longer periods (>17 months) were also identified, although their periodicity may be less reliable due to the limited number of time points. Many genes with annual periodicity were shared across all species, including 1280 genes in leaves and 1894 genes in buds (*Figure 3—figure supplement 1B*). In contrast, most genes exhibiting half-annual periodicity were species-specific, with no genes shared among all the species (*Figure 3—figure supplement 1C*).

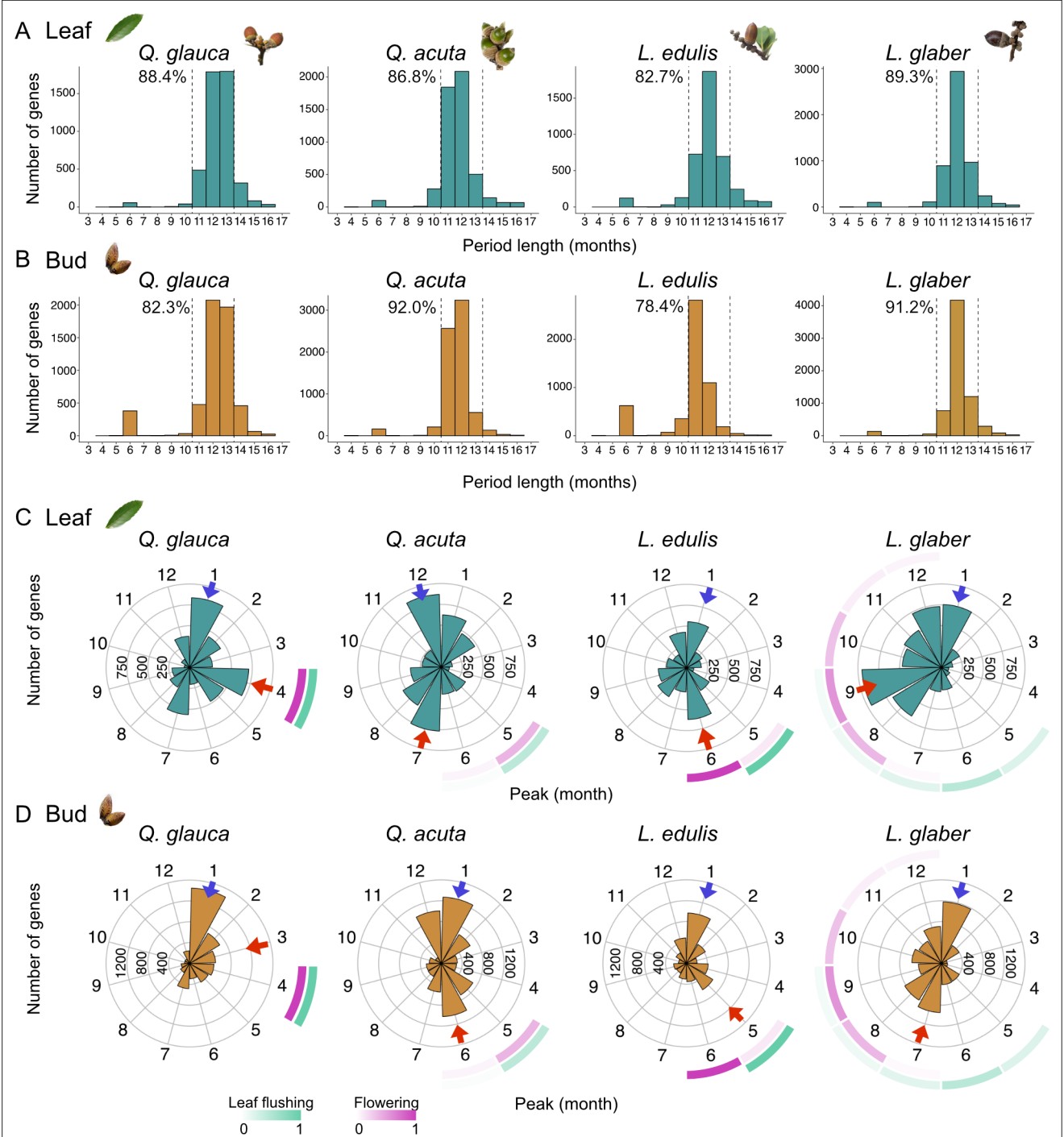

**Figure 3.** Distribution of period and peak month in rhythmic gene expression. (**A, B**) Distribution of period length (month) among the genes with significant rhythmicity calculated by RAIN for leaf (**A**) and bud (**B**) samples of each species. (**C, D**) Distribution of seasonal peaks among the genes with annual periodicity. The angle of each bar represents the peak month, while the bar height indicates the number of genes. Blue arrows indicate the month with the highest number of genes peaking during winter (December–February), while red arrows indicate the month with the highest number of genes peaking during the growing season (March–November). Purple and green squares around each rose plot denote the observed phenology of flowering and leaf flushing (proportion), respectively.

The online version of this article includes the following figure supplement(s) for figure 3:

**Figure supplement 1.** Number of rhythmic genes across species and distribution of peak months across half-annual genes.

## Peak month distribution of rhythmic genes and intra-genus and inter-genera comparison

To determine the seasonal timing of gene expression peaks, we focused on genes with annual or half-annual periodicity and visualized the density and direction of these peaks throughout the year using a circular histogram (*Figure 3C and D*; *Supplementary file 4*). On average, 34.9% and 41.3% of annual rhythmic genes peaked in winter (December–February) in leaf and bud samples, respectively. Additionally, another peak corresponding to or occurring slightly after leaf flushing and flowering events was observed, with its timing varying by species: April in *Q. glauca* (15.6%), July in *Q. acuta* (15.3%), June in *L. edulis* (16.0%), September in *L. glaber* (18.1%), for leaf tissues (*Figure 3C*). Similar peaks corresponding to leaf flushing or flowering were also detected in bud samples, occurring approximately one to two months earlier than in leaf samples (*Figure 3D*). For genes with half-annual periodicity, the majority (63.2±4.8%) exhibited expression peaks both in spring (March–May) and autumn (September–November) in both tissues (*Figure 3—figure supplement 1D*; *Supplementary file 4*), with the exception of *L. glaber*, an autumn-flowering species. Interestingly, *WRKY22*, a transcription factor regulating ethylene biosynthesis and senescence (*Zhu et al., 2024*), exhibited half-annual periodicity in *L. glaber* (*Figure 3—figure supplement 1E*). In addition, *SNF1-RELATED PROTEIN KINASE REGULATORY SUBUNIT GAMMA 1* (*KINγ1*), a subunit of SNF1-related kinase (*SnRK*) involved in stress response such as starvation (*Emanuelle et al., 2016*), showed two distinct peaks in *Q. glauca* and *L. edulis*, occurring in both summer (July) and winter (January) (*Figure 3—figure supplement 1E*). This result suggests that both summer and winter pose stressful conditions for these species.

To compare the peak month of gene expression with annual periodicity across species, we quantified the number of genes within each peak combination for both intra-genus (*Figure 4A*) and inter-genera species pairs in buds (*Figure 4B*) and leaves (*Figure 4—figure supplement 1A*, B). During winter, a large number of genes exhibited highly synchronized expression peaks across species, with high pairwise Pearson's correlation coefficients observed (*Figure 4A and B*). In contrast, the peak timing of gene expression from spring to autumn varied across species, aligning with differences in leaf flushing and flowering times (*Figure 4A and B*). Some genes exhibited pronounced divergence, showing negative Pearson's correlation coefficients (<–0.3) between species. Notably, the number of such genes was larger in inter-genera comparisons (11–77 genes; *Figure 4B*) than in intra-genus comparisons (4–12 genes; *Figure 4A*). A similar pattern was observed in leaf samples (*Figure 4—figure supplement 1A, B*). When comparing seasonal expression peaks between tissues, the majority of genes were synchronously activated in winter in both tissues (*Figure 4—figure supplement 1C*). During spring to summer, slight variations in seasonal expression peaks were observed among tissues (*Figure 4—figure supplement 1C*).

We quantified the degree of divergence in seasonal gene expression patterns across species by calculating the peak month differences between two species, where one species exhibits an expression peak in the given month. We defined the molecular phenology divergence index (*D*) for each month by quantifying the proportion of genes with peak month differences greater than 2 months between two species, relative to the total number of genes with expression peaks in that month. From spring to autumn, the mean of *D* over all species pairs in buds ranged from 0.47 to 0.66, but it declined sharply during winter, reaching a minimum of 0.12 in January in buds (Nemenyi test, $p<0.05$; *Figure 4C*). A similar trend was observed in leaf samples, with another trough in summer (*Figure 4—figure supplement 1D*). This seasonal trend remained consistent even after excluding *Q. acuta*, which was sampled from a slightly cooler environment than the other three species, from the analysis (*Figure 4—figure supplement 2*), indicating that the observed patterns are not driven by environmental differences associated with elevation. The values of *D* were not consistently smaller in inter-genera pairs compared to intra-genus pairs. These findings suggest that there is a seasonal constraint in gene expression in which the degree of molecular phenology divergence is not constant but varies significantly across seasons, with the least divergence occurring in winter.

## Phylogenetic constraints in the evolution of seasonal gene expression

We next investigated whether phylogenetic relationship influences the evolution of seasonal gene expression by calculating pairwise Pearson's correlations of seasonal expression for each gene across all species pairs (*Supplementary file 5*). The median of Pearson's correlation coefficients across all genes were significantly higher between species within the same genus compared to those between

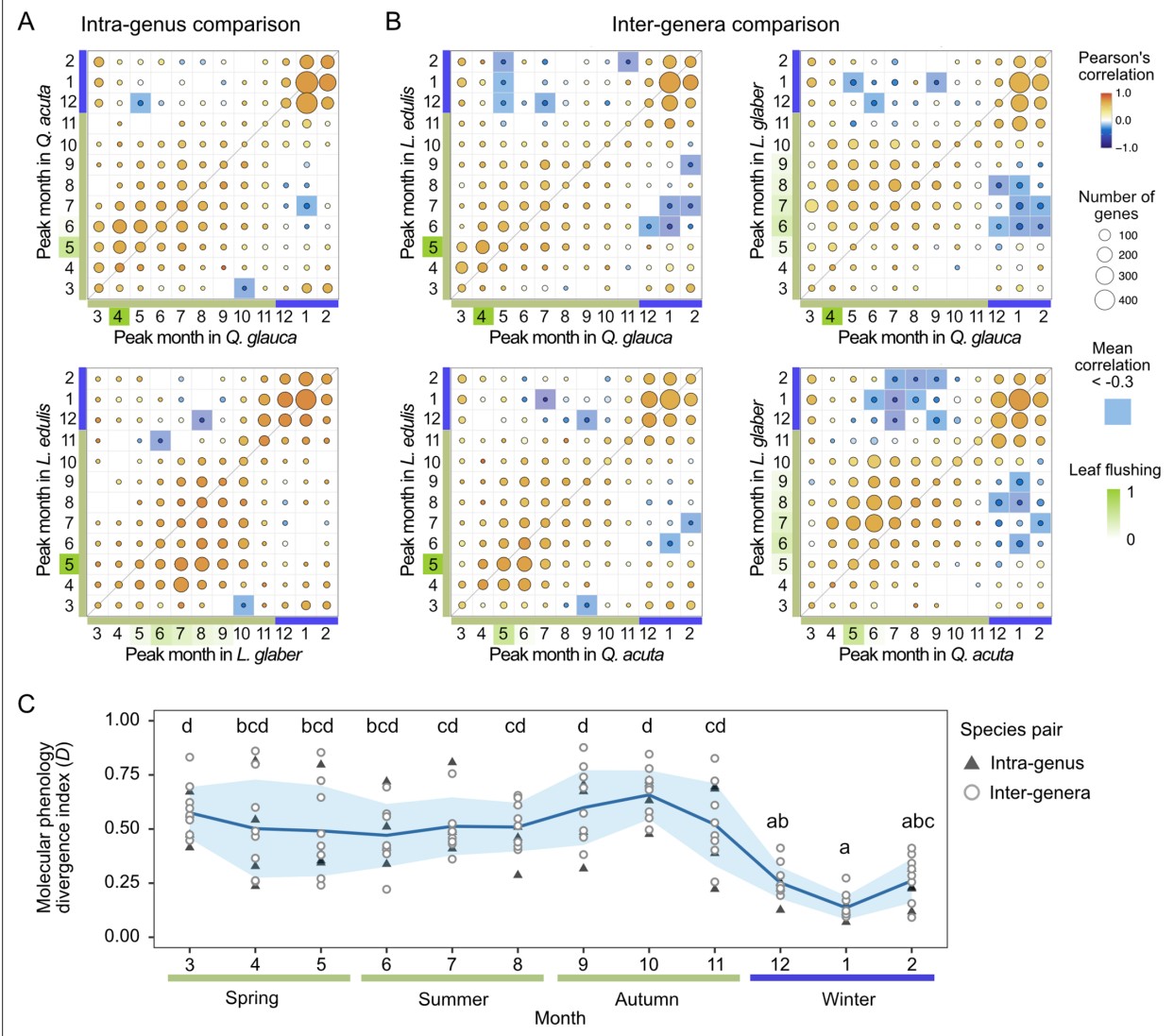

**Figure 4.** Comparison of peak months in seasonal gene expression across species in buds. (**A**) Intra-genus and (**B**) inter-genera comparisons of peak months in seasonal gene expression. The size and color of the circles represent the number of genes and the mean of Pearson's correlation, respectively. Genes with negative correlations lower than –0.3 are highlighted in blue, while the month of leaf flushing is highlighted in green. (**C**) Plot of molecular phenology divergence index $D$ across months. Triangles and circles indicate the values of $D$ calculated from intra-genus and inter-genera comparison, respectively. The line and shaded envelope indicate the mean and standard deviation (s.d.). Different letters denote statistically significant differences (Nemenyi test, $p<0.05$).

The online version of this article includes the following figure supplement(s) for figure 4:

**Figure supplement 1.** Comparison of peak months in seasonal gene expression across species in leaves.

**Figure supplement 2.** Seasonal difference in molecular phenology divergence index $D$ calculated for species under the identical environmental condition.

species from different genera in both leaves (Steel-Dwass test, $p<0.05$; *Figure 5—figure supplement 1A*; *Figure 5—figure supplement 2*) and buds (Steel-Dwass test, $p<0.05$; *Figure 5A*; *Figure 5—figure supplement 2*). The intra-genus comparison in genus *Lithocarpus* exhibited the highest correlation in their seasonal gene expression, despite their notable differences in flowering phenology, with *L. edulis* flowering in spring and *L. glaber* in autumn. In contrast, the intra-genus comparison in genus *Quercus* resulted in a relatively low correlation in their gene expression, regardless of relatively similar phenology in leaf flushing and flowering. This difference may be attributed to environmental factors, as *Q. acuta* is the only species found in the high-altitude study site with colder climates compared to the other three species.

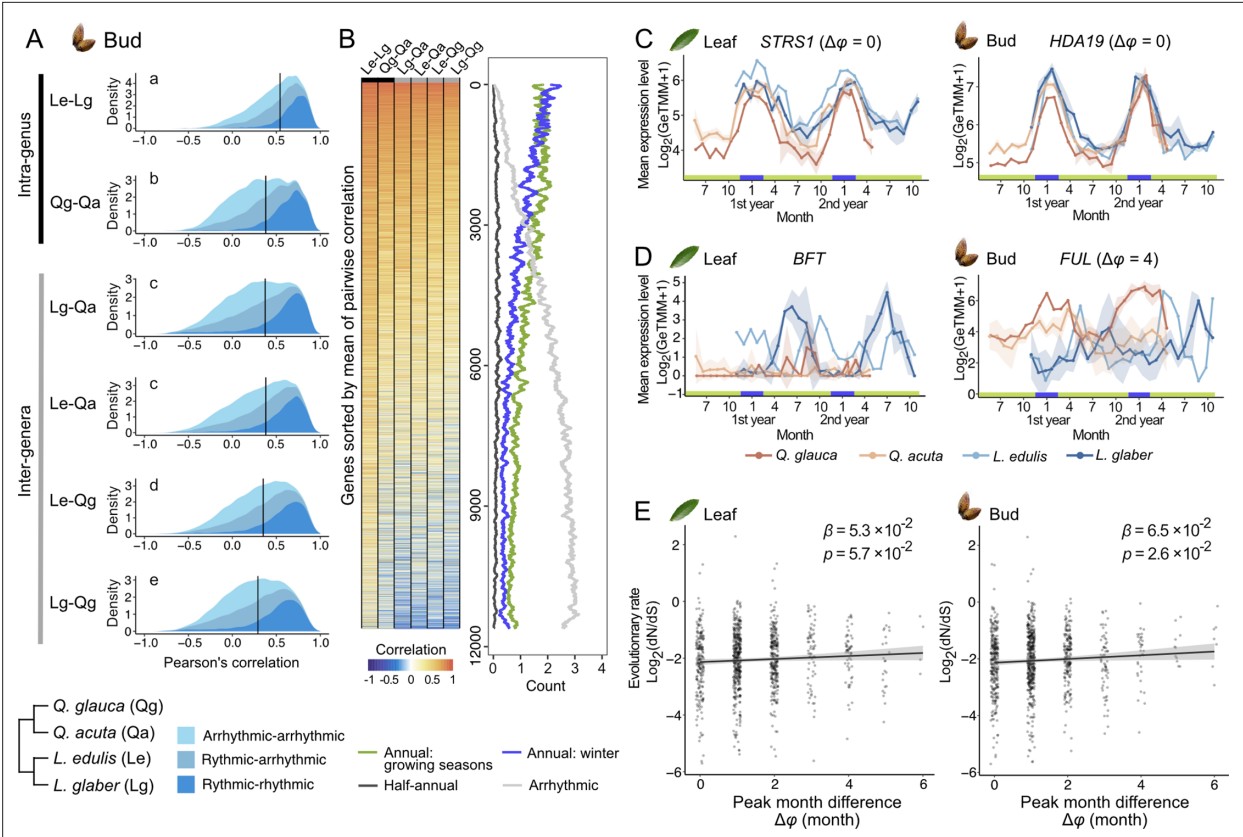

**Figure 5.** Phylogenetic constraints in the evolution of seasonal gene expression and the relationship between seasonal gene expression divergence and sequence evolution. (**A**) Distribution of pairwise Pearson's correlation coefficients for gene expression across species (Le: *L. edulis*, Lg: *L. glaber*, Qg: *Q. glauca*, Qa: *Q. acuta*; n=11,749). Colors indicate rhythmicity categories. Black bars show median correlation values, and different letters indicate significant differences (Nemenyi test, p<0.05). (**B**) Heatmap of pairwise correlation coefficients for each gene, with mean gene counts in four seasonal expression categories: (1) annual rhythmic genes peaking in the growing season (March–November), (2) annual rhythmic genes peaking in winter (December–February), (3) half-annual rhythmic genes, and (4) arrhythmic genes. Genes are ordered by mean correlation across species pairs. (**C, D**) Conserved (**C**) and diverged (**D**) gene expression patterns, based on the top and bottom 5% of pairwise correlations. Solid lines and error bars indicate mean ± SD. Numbers in parentheses show peak month differences between *Q. glauca* and *L. edulis*. Peak month difference for *BROTHER OF FT AND TFL1* (*BFT*) was not calculated due to arrhythmic expression in *Q. glauca*. (**E**) Relationship between evolutionary rate (dN/dS) and peak month difference ($\Delta\varphi$) for genes with annual periodicity in *Q. glauca* and *L. edulis* (n=951). Black lines and transparent bands represent regression lines and 95% confidence intervals. The regression coefficient ($\beta$) and corresponding p-value are shown in each panel.

The online version of this article includes the following figure supplement(s) for figure 5:

**Figure supplement 1.** Phylogenetic constraints in the evolution of seasonal gene expression in leaves.

**Figure supplement 2.** Distribution of pairwise Pearson's correlation coefficients for gene expression across species under the identical environmental conditions.

**Figure supplement 3.** Pairwise mean absolute difference of gene expression dynamics.

**Figure supplement 4.** Gene Ontology (GO) enrichment analysis.

**Figure supplement 5.** Relationship between sequence divergence and gene expression divergence.

**Figure supplement 6.** Developmental constraints and seasonal constraints in the evolution of gene expression.

Rhythmic genes exhibited more conserved expression patterns than arrhythmic genes in both tissues (*Figure 5A*; *Figure 5—figure supplement 1A*). In particular, genes with seasonal expression peaks in winter showed high correlation across species in both tissues (*Figure 5B*; *Figure 5—figure supplement 1B*). The proportion of rhythmic genes with annual periodicity declined as the mean correlation coefficient across species decreased, with a more rapid decline observed for genes peaking in winter compared to those with peak expression during the growing seasons (*Figure 5B*; *Figure 5—figure supplement 1B*). Additionally, the mean absolute difference in seasonal gene expression dynamics was significantly smaller between species within the same genus than between

species from different genera in both tissues (*Figure 5—figure supplement 3A, B*). The relationship between the mean absolute difference and the proportion of rhythmic genes was less clear compared to that of the mean correlation coefficient (*Figure 5—figure supplement 3C, D*). These findings suggest that, in addition to seasonal constraints, the overall dynamics of seasonal gene expression are also shaped by phylogenetic constraints.

## Functional annotation of genes with conserved or divergent expression across species

To functionally annotate genes with conserved or divergent expression across species, we extracted genes with the top and bottom 5% based on the distribution of mean correlation (*Figure 5B*; *Figure 5—figure supplement 1B*) and conducted Gene Ontology (GO) enrichment analysis. GO analysis revealed that the top 5% of genes in leaves were associated with photosynthesis, while those in buds were linked to cell fate determination (*Figure 5—figure supplement 4A*; *Supplementary file 6*). The gene with the highest correlation in leaves was *STRESS RESPONSE SUPPRESSOR1* (*STRS1*; mean of pairwise correlation = 0.91; *Figure 5C*), a DEAD-box RNA helicase previously reported to enhance tolerance to abiotic stresses such as cold (*Kant et al., 2007*). In buds, the most highly correlated gene was the histone deacetylase *HDA19* (mean of pairwise correlation = 0.93; *Figure 5C*), which plays a key role in environmental stress responses through histone modification and chromatin remodeling (*Chen and Wu, 2010*). These findings suggest that genes associated with photosynthesis during the growing season and stress responses during winter exhibit highly conserved seasonal expression patterns across species.

The bottom 5% of genes were predominantly associated with DNA replication and cell division in leaves (*Figure 5—figure supplement 4B*), aligning with the divergent leaf phenology observed (*Figure 3B*). In contrast, GO terms related to immune response were weakly enriched in buds (*Figure 5—figure supplement 4B*). Notably, we identified MADS-box genes linked to floral transition among the bottom 5% of genes: *BROTHER OF FT AND TFL1* (*BFT*) in leaves (mean of pairwise correlation = –0.094) and *FRUITFULL* (*FUL*) in buds (mean of pairwise correlation = –0.069). *BFT*, a floral repressor (*Yoo et al., 2010*), was expressed in autumn in the spring-flowering species *L. edulis* and in spring in the autumn-flowering species *L. glaber* leaves (*Figure 5D*). Meanwhile, *FUL*, a positive regulator of floral transition (*Smaczniak et al., 2012*), was expressed in spring in *L. edulis* and in autumn in *L. glaber* (*Figure 5D*), mirroring the timing of flowering. In *Q. glauca* and *Q. acuta*, *FUL* was expressed in winter (*Figure 5D*), nearly out of phase with their expressions in *Lithocarpus* species. These results suggest that genes involved in reproductive processes exhibit species-specific divergence in expression patterns in response to seasonal environmental changes.

## Relating seasonal gene expression divergence to sequence divergence

If divergence of seasonal timing of gene expression is related to selective constraint, we would expect a correlation between sequence divergence patterns and the divergence in seasonal expression peaks. To explore the relationship between the divergence of seasonal gene expression and protein sequence evolution, we calculated synonymous (dS) and non-synonymous substitution rates (dN) and their ratio (dN/dS) for each gene using coding sequences from newly assembled genomes of *Q. glauca* and *L. edulis*. We first confirmed that the highly expressed genes evolve more slowly by identifying a significant negative correlation between evolutionary rate (dN/dS) and mean of the gene expression ($r$=–0.21, $p$=1.8×10$^{-118}$ for dN/dS; *Figure 5—figure supplement 5A*; *Supplementary file 7*). This finding aligns with previous studies in yeasts, mammals, and plants (*Drummond et al., 2005*; *Gaut et al., 2011*; *Liao and Zhang, 2006*; *Yang and Gaut, 2011*). By targeting the genes that exhibited annual periodicity across both species and tissues, we examined the relationship between these evolutionary measures and peak month differences in seasonal gene expression between species ($\Delta\varphi$) in leaves and bud tissues using linear regression. Assessment of the significance of the regression coefficient using a $t$-test revealed that genes with smaller peak month differences between species exhibited lower values of dN/dS in buds ($p$=2.6×10$^{-2}$) and marginally lower values of dN/dS in leaves ($p$=5.7×10$^{-2}$). The lower values were also found in dN in both leaves ($p$=8.1×10$^{-3}$) and buds ($p$=7.8×10$^{-3}$), whereas no significant trend was observed for dS in both tissues (*Figure 5E*; *Figure 5—figure supplement 5B*, C; *Supplementary file 7*). We further investigated whether the association between the divergence in seasonal expression peaks and sequence divergence patterns

varies between genes expressed in winter and in the growing season. We classified genes that exhibited annual periodicity across both species and tissues into four groups based on their expression peak timing (winter or growing seasons) and tissue specificity (leaves or buds) and compared the extent of protein sequence divergence across these groups. We found no significant differences in dN/dS (Kruskal-Wallis test, $p=2.8\times10^{-1}$; *Figure 5—figure supplement 5D*; *Supplementary file 7*) between genes expressed in winter in both tissues and those in growing seasons. The same result was observed even when focusing only on genes with conserved peaks (peak month difference <2), with no significant differences in dN/dS between these gene sets (Kruskal-Wallis test, $p=3.5\times10^{-1}$; *Figure 5—figure supplement 5E*; *Supplementary file 7*). These results suggest that coding-sequence variation is unlikely to be the primary driver of seasonal gene expression patterns.

## Discussion

A central goal in biology is to unravel the molecular basis of phenotypic evolution, with gene expression serving as a fundamental driver of this process. In this study, we investigated the impact of seasonal environmental fluctuations on the evolution of gene expression and its relationship with phenological traits in four dominant *Fagaceae* tree species in forest ecosystems. We found that gene expression profiles were more resistant to change during winter compared to the growing season, indicating the presence of a seasonal constraint on the evolution of gene expression. This constraint becomes particularly pronounced in buds during winter when temperatures fall below approximately 10 °C (*Figure 2B*). This temperature threshold aligns with previous findings, where winter-specific transcript profiles were reported at temperatures below 8.7–11.3°C in cherry trees (*Miyawaki-Kuwakado et al., 2024*), 11.3 °C in evergreen *Fagaceae* (*Satake et al., 2023*), and 10.5 °C in perennial *Arabidopsis* (*Aikawa et al., 2010*). These results suggest the presence of a conserved mechanism for gene expression regulation under low-temperature environments across different species. The greater conservation of gene expression patterns in buds compared to leaves across species may be attributed to their functional role protecting essential meristematic tissues, including the shoot apex and leaf and flower primordia. Additionally, buds likely play a crucial role in maintaining common physiological functions necessary for the resumption of growth in spring. This functional consistency may impose evolutionary constraints on gene expression, contributing to its conservation across species.

It is intriguing to compare the seasonal constraints identified in this study with developmental constraints in gene expression evolution, where evolutionary conservation and variation across developmental stages during the life cycle have been observed, such as in the early conservation model and the developmental hourglass model (*Figure 5—figure supplement 6A*; *Irie and Kuratani, 2011*; *Lotharukpong et al., 2024*; *Quint et al., 2012*). Previous studies have demonstrated that certain developmental stages exhibit remarkably similar gene expression patterns (*Drost et al., 2016*; *Kalinka et al., 2010*; *Levin et al., 2016*). Our findings highlight that such highly similar gene expression patterns are observed during the winter season (*Figure 5—figure supplement 6B*), suggesting that a common physiological state is essential for enduring harsh winter conditions. This seasonal constraint on gene expression evolution may have the potential to limit temporal niche partitioning at the phenotypic level and slow species divergence rates in seasonal environments (*Cleland and Wolkovich, 2024*; *Kubo and Iwasa, 1996*). Although completely disentangling the effects of developmental stages and seasonal changes on gene expression is challenging when using molecular phenology data collected from natural environments, the observation that species with different phenology—i.e., different developmental timing in seasonal environments—exhibit convergent gene expression patterns during winter underscores the profound influence of winter environmental conditions on the evolution of gene expression. Future research is needed to elucidate the mechanisms driving the establishment of such conserved gene expression patterns, including the dynamic reorganization of chromatin architecture in response to cold (*Fischl et al., 2020*; *Nishio et al., 2020*; *Zhang et al., 2023*).

We found that the evolution of coding sequences was only weakly correlated with the divergence of seasonal peaks in gene expression across species. To understand the evolutionary mechanism underlying the seasonal constraint on gene expression that results in similar expression patterns during winter, an important future direction is to explore the evolution of non-coding sequences that modulate gene expression levels in response to seasonal environmental changes. Regulatory variation has been recognized as a major driver of evolutionary innovation (*Prud'homme et al., 2007*; *True and*

*Haag, 2001*). Recent deep learning studies utilizing genome and transcriptome data from yeast and humans have highlighted the significance of co-evolution between coding and non-coding sequences in shaping gene expression levels (*Zrimec et al., 2021*; *Zrimec et al., 2020*). Understanding how seasonal environments drive the co-evolution of coding and regulatory sequences, as well as the divergence of molecular phenology and its phenological outputs, will require expanding analyses to a broader range of species. Such efforts will facilitate the development of predictive models for evolutionary processes influenced by seasonal dynamics. Comparison of molecular phenology across wide climatic zones, including both tropical and temperate zones, will also be effective to identify key genes for the phenological response to ongoing climate change.

## Materials and methods

**Key resources table**

| Reagent type (species) or resource | Designation | Source or reference | Identifiers | Additional information |
|---|---|---|---|---|
| Biological sample (*Quercus glauca*) | Leaf | This paper | | Freshly isolated from *Q. glauca* |
| Biological sample (*Q. glauca*) | Bud | This paper | | Freshly isolated from *Q. glauca* |
| Biological sample (*Quercus acuta*) | Leaf | This paper | | Freshly isolated from *Q. acuta* |
| Biological sample (*Q. acuta*) | Bud | This paper | | Freshly isolated from *Q. acuta* |
| Biological sample (*Lithocarpus edulis*) | Leaf | This paper | | Freshly isolated from *L. edulis* |
| Biological sample (*L. edulis*) | Bud | This paper | | Freshly isolated from *L. edulis* |
| Biological sample (*Lithocarpus glaber*) | Leaf | This paper | | Freshly isolated from *L. glaber* |
| Biological sample (*L. glaber*) | Bud | This paper | | Freshly isolated from *L. glaber* |
| Commercial assay or kit | QIAamp DNA Mini Kit | Qiagen | Cat. #: 51304 | |
| Commercial assay or kit | SMRTbell Express Template Prep Kit 2.0 | PacBio | Cat. #: 100-938-900 | |
| Commercial assay or kit | Sequel II Binding Kit 2.0 | PacBio | Cat. #: 101-789-500 | |
| Commercial assay or kit | PureLink Plant RNA Reagent | Thermo Fisher Scientific | Cat. #: 12322012 | |
| Commercial assay or kit | TURBO DNA-free kit | Thermo Fisher Scientific | Cat. #: AM1907 | |
| Commercial assay or kit | NEBNext Ultra II RNA Library Prep Kit for Illumina | New England BioLabs | Cat. #: E7770 | |
| Commercial assay or kit | NEBNext Ultra II DNA Library Prep Kit | New England BioLabs | Cat. #: E7645 | |
| Commercial assay or kit | NEBNext Multiplex Oligos for Illumina | New England BioLabs | Cat. #: E7335, E7500, E7710 | |
| Commercial assay or kit | RNeasy Plant Mini Kit | Qiagen | Cat. #: 74904 | |
| Commercial assay or kit | Agilent RNA 6000 Nano kit | Agilent Technologies | Cat. #: 5067–1511 | |
| Commercial assay or kit | High Sensitivity DNA kit | Agilent Technologies | Cat. #: 5067–4626 | |
| Commercial assay or kit | NEBNext Library Quant Kit for Illumina | New England BioLabs | Cat. #: E7630 | |
| Chemical compound, drug | RNA-stabilizing reagent (RNAlater) | Thermo Fisher Scientific | Cat. #: AM7021 | |
| Software, algorithm | Hifiasm1 | Hifiasm | Hifiasm1 v0.15.4-r347 | |

*Continued on next page*

*Continued*

| Reagent type (species) or resource | Designation | Source or reference | Identifiers | Additional information |
|---|---|---|---|---|
| Software, algorithm | blobtools2 | BlobToolKit | RRID:SCR_023351 | |
| Software, algorithm | purge_dups3 | purge_dups | RRID:SCR_021173 | |
| Software, algorithm | HiRise | *Putnam et al., 2016* | RRID:SCR_023037 | |
| Software, algorithm | fastp | *Chen et al., 2018* | RRID:SCR_016962 | |
| Software, algorithm | STAR | *Dobin et al., 2013* | RRID:SCR_004463 | |
| Software, algorithm | edgeR | edgeR | RRID:SCR_012802 | |
| Software, algorithm | Braker2 | *Brůna et al., 2021* | RRID:SCR_018964 | |
| Software, algorithm | GeMoMa | *Keilwagen et al., 2019* | RRID:SCR_017646 | |
| Software, algorithm | GFFcompare | *Pertea and Pertea, 2020* | | |
| Software, algorithm | DIAMOND | *Buchfink et al., 2015* | RRID:SCR_016071 | |
| Software, algorithm | Salmon | *Patro et al., 2017* | RRID:SCR_017036 | |
| Software, algorithm | eggNOG-mapper | *Cantalapiedra et al., 2021* | RRID:SCR_021165 | |
| Software, algorithm | BLASTp | *Camacho et al., 2009* | RRID:SCR_001010 | |
| Software, algorithm | MCScanX | *Wang et al., 2012* | RRID:SCR_022067 | |
| Software, algorithm | GenomeScope2 | *Ranallo-Benavidez et al., 2020* | | |
| Software, algorithm | KmerGenie | *Chikhi and Medvedev, 2014* | | |
| Software, algorithm | OrthoFinder2 | *Emms and Kelly, 2019* | RRID:SCR_017118 | |
| Software, algorithm | R | R project | RRID:SCR_001905 | |
| Software, algorithm | PAML | *Yang, 2007* | RRID:SCR_014932 | |

## Sample collection for genome sequencing

To construct genomes of *Q. glauca* and *L. edulis*, we collected approximately 5 g of fresh leaves from one individual in spring, respectively. We shipped these samples to Dovetail Genomics, LLC (Scotts Valley, CA, USA), where Dovetail staff performed DNA extraction, library preparation, sequencing, and assembly steps.

## Genome sequencing and assembly

DNA extraction was performed using the QIAamp DNA Mini Kit (Qiagen, Germantown, MD, USA) using the manufacturer's recommended protocol. Mean fragment length of the extracted DNA was 60 kbp for *Q. glauca* and 65 kbp for *L. edulis*. DNA samples were quantified using Qubit 2.0 Fluoro-meter (Life Technologies, Carlsbad, CA, USA). The library preparation, sequencing, and scaffolding were carried out by Dovetail Genomics (California, USA) according to their standard genome assembly pipeline (https://cantatabio.com/genome-assembly-services). The PacBio SMRTbell library (~20 kb) for PacBio Sequel was constructed using SMRTbell Express Template Prep Kit 2.0 (PacBio, Menlo Park, CA, USA) using the manufacturer's recommended protocol. The library was bound to polymerase using the Sequel II Binding Kit 2.0 (PacBio) and loaded onto PacBio Sequel II. Sequencing was performed on PacBio Sequel II 8 M SMRT cells. PacBio CCS reads were used as an input to Hifiasm1 v0.15.4-r347 with default parameters. For *Q. glauca*, we also generated sequence data using PacBio Revio. BLASTn results of the contigs obtained by Hifiasm assembly against the NCBI's NT database were used as input for blobtools2 v1.1.1 and contigs identified as possible contamination were removed from the assembly. Finally, purge_dups3 v1.2.5 was used to remove haplotigs and contig overlaps.

## Assembly scaffolding with HiRise

A proximity ligation library was generated by the Dovetail's Omni-C technique, followed by sequencing on an Illumina HiSeqX platform. Chromatin was fixed in place in the nucleus with formaldehyde before

extraction (https://cantatabio.com/omni-c). Fixed chromatin was digested with DNase I, and the fragmented chromatin ends were repaired and biotinylated to adapters followed by proximity ligation. Crosslinks were then reversed, the DNA purified, and the biotin subsequently removed. The DNA library was prepared and sequenced to produce 2×150 bp paired-end reads (around 30x coverage). The Omni-C technology uses a sequence-independent endonuclease which provides even, unbiased genome coverage. The input de novo assembly and Omni-C library reads were used as input data for HiRise, a software pipeline designed specifically for using proximity ligation data to scaffold genome assemblies (*Putnam et al., 2016*). The Omni-C library sequences were aligned to the contigs assembled by Hifiasm using bwa (*Li and Durbin, 2009*). The separations of Omni-C reads from paired-end mapped within the contigs were analyzed by HiRise to produce a likelihood model for genomic distance between paired-end reads, and the model was used to identify and break putative misjoins, to score prospective joins, and make joins above a threshold.

## Sample collection for RNA sequencing

To monitor the genome-wide gene expression profiles, we collected leaf, bud, and flower samples from four Fagaceae species, *Q. glauca*, *Q. acuta*, *L. edulis,* and *L. glaber*, every four weeks for two years (October 2019 to October 2021 for the *Lithocarpus* species and April 2021 to April 2023 for the *Quercus species*) across Fukuoka and Saga prefectures in Kyushu, Japan. We targeted three individuals, and each sample was collected from three branches per individual, except for *L. edulis*. Given the consistent expression patterns observed among individuals in a previous study (*Satake et al., 2023*), we focused on only one individual for *L. edulis*. Study sites for *Q. glauca* is the biodiversity reserve on the Ito campus of Kyushu University (33°35′ N, 130°12′ E, 20–57 m a.s.l). *L. edulis* and *L. glaber* were studied at the Imajuku Field Activity Center (33°33′ N, 130°16′ E, 84–111 m a.s.l). The Imajuku and Ito Campus study sites, located only 7.3 km apart, experience nearly identical temperature conditions. *Q. acuta* was monitored in the Sefuri mountains (33°26′ N, 130°22′ E, 970–974 m a.s.l), a higher-altitude, cooler site. Environmental data from Ito Campus and the Imajuku Field Activity Center were obtained from the Japan Meteorological Agency, while data from Sefuri Mountain were collected using a logger installed at the sampling site. Additionally, to assess the potential impact of different sampling periods between the *Lithocarpus* and *Quercus* species on the gene expression analysis, we conducted an additional year of sampling for *L. edulis* and *L. glaber*, collecting leaf and bud samples from November 2021 to November 2022, and used the samples collected within the same period for all the four species (May 2021 to November 2022).

Samples for RNA-seq were taken from the sun-exposed crown (approximately 2–5 m from the ground) using long pruning shears from 11:30 to 12:30 h. For each sample, 0.2–0.4 g of tissue was preserved in a 2 ml microtube containing 1.5 ml of RNA-stabilizing reagent (RNAlater; Thermo Fisher Scientific, Waltham, MA, USA) immediately after harvesting. The samples were transferred to the laboratory within 2 hr after sampling, stored at 4 °C overnight and then stored at −80 °C until RNA extraction. During transport to the laboratory, the samples were kept in a cooler box with ice to maintain a low temperature.

## RNA extraction and RNA-seq analysis

RNA was extracted independently from bud and leaf samples of each tree. Total RNA of the leaf sample was extracted in accordance with the method described in a previous study (*Miyazaki et al., 2014*). Total RNA of the bud sample was extracted using the PureLink Plant RNA Reagent (Thermo Fisher Scientific, Waltham, MA, USA) and subsequently processed with DNase using the TURBO DNA-free kit (Thermo Fisher Scientific) according to their protocols and purified using the RNeasy Plant Mini Kit (Qiagen, Hilden, Germany) according to its RNA cleanup protocol. RNA integrity was examined using the Agilent RNA 6000 Nano kit on a 2100 Bioanalyzer (Agilent Technologies), and the RNA yield was determined using a NanoDrop ND-2000 spectrophotometer (Thermo Fisher Scientific). Five to six micrograms of RNA extracted from each sample were sent to Hangzhou Veritas Genetics Medical Institute Co., Ltd., where a cDNA library was prepared with a NEBNext Ultra II RNA Library Prep Kit for Illumina, and 150 base paired-end transcriptome sequencing of each sample was conducted using an Illumina NovaSeq6000 sequencer (Illumina).

The quantification of gene expression followed a three-step process. First, quality filtering was performed using the fastp v0.23.2 with default options: phred quality ≥15, the maximum number of

N base in one read = 5 (*Chen et al., 2018*). Second, RNA-seq reads were mapped to the reference genome using STAR v2.7.10b (*Dobin et al., 2013*). Third, RNA-seq reads were quantified using RSEM v1.3.1 (*Li and Dewey, 2011*). We excluded the genes with low expression level using the threshold, the value of mean Reads Per Kilobase (RPK)<1 for each species. Then, we further normalized the expression level using the value of Gene length corrected Trimmed Mean of M-values (GeTMM) (*Smid et al., 2018*) using an R package 'edgeR' v3.42.4. For the downstream analysis, we calculated the mean expression levels of GeTMM of each gene across multiple individuals and transformed them to $\log_2(\text{GeTMM} +1)$. To confirm that the effect of the difference in sampling period is negligible, we added the RNA-Seq data of *L. edulis* and *L. glaber* collected from December 2021 to December 2022. We extracted the same genes in the existing data that were curated based on the mean RPK values across two-year sampling periods. Then we further normalized the expression level using GeTMM.

## Genome annotation

The RNA-seq reads of *Q. glauca* (8 samples obtained by a previous study (20) collected from May to December 2017; GEO's accession number: GSE211384) and *L. edulis* (20 samples covering leaf, bud, and flower tissues collected in January, April, July, and November 2021) were, respectively, mapped against the scaffolds of *Q. glauca* and *L. edulis* obtained by HiRise with HISAT2 v2.1.0 (*Kim et al., 2019*). The bam files generated by the mappings and amino acid sequences of the 9 species obtained from NCBI (*Castanea mollissima*: GCA_000763605.2 (NCBI's accession number), *Q. lobata*: GCF_001633185.1, *Q. robur*: GCF_932294415.1, *Q. suber*: GCF_002906115.1, *Carpinus fangiana*: GCA_006937295.1, *Carya illinoinensis*: GCF_018687715.1, *Juglans macrocarpa×J. regia*: GCF_004785595.1, *Juglans regia*: GCF_001411555.2, *Morella rubra*: GCA_003952965.2) were used for ab initio gene prediction by Braker2 v2.1.6 (*Brůna et al., 2021*), and the genes with highest scores were selected to exclude splicing variants. In parallel, the gene models were, respectively, constructed in *Q. glauca* and *L. edulis* by GeMoMa v1.7.1 (*Keilwagen et al., 2019*) with RNA-seq reads and amino acid sequences used for the gene prediction in Braker2, and the genes with highest scores were selected to exclude splicing variants. The duplicated genes between Braker2 and GeMoMa were excluded by GFFcompare v0.11.2 (*Pertea and Pertea, 2020*). The remaining genes were searched against UniProtKB (*Bateman et al., 2025*) by DIAMOND v2.0.5 (*Buchfink et al., 2015*) in the 'more-sensitive' option with E-value≤$10^{-20}$ and identity ≥25%. Next, gene expression analysis was performed by Salmon v1.4.0 (*Patro et al., 2017*), and the genes with TPM values >0.1 were considered as expressed. The orthologous genes were searched against EggNOG (*Hernández-Plaza et al., 2023*) using eggNOG-mapper v2.1.8 (*Cantalapiedra et al., 2021*) with E-value≤$10^{-10}$. The genes were searched against protein sequences of *Castanea mollissima* (Accession: GCA_000763605.2), *Quercus lobata* (GCF_001633185.2), and *Arabidopsis thaliana* (Araport11 *Krishnakumar et al., 2015* by BLASTp v2.13.0) (*Camacho et al., 2009*) with E-value≤$10^{-20}$. According to these similarity searches, the genes whose product names contain keywords related to transposable elements (TEs) were categorized into TE. The genes having similarity hits and being expressed by Salmon were classified into high-confidence (HC) genes, while the remaining genes were categorized into low-confidence (LC) genes. The genes categorized into HC were used for further analyses.

## Genome synteny analysis

To investigate the syntenic relationship between *Q. glauca* and *L. edulis*, we first performed an all-vs-all BLASTp search of the protein sequences, using a cut-off E-value of 10−5. Collinearity blocks within and between the genomes were then identified using MCScanX (*Wang et al., 2012*) based on the results of the BLASTp searches. Finally, the collinearity blocks were visualized using SynVisio (https://synvisio.github.io).

## Illumina short-read sequencing and library preparation

For genome size estimation and phylogenetic reconstruction, we also generated short-read sequencing of *Q. glauca*, *Q. acuta*, *L. edulis*, and *L. glaber*. DNA extraction was performed using a modified version of the method described previously (*Satake et al., 2024b*). The DNA samples were sent to Macrogen Inc (Republic of Korea) for sequencing on the Illumina HiSeqX platform (Illumina, San Diego, CA, USA). DNA was sheared to around 500 bp fragments in size using dsDNA fragmentase (New England BioLabs, Ipswich, MA, USA). Library preparation was performed using the NEBNext

Ultra II DNA Library Prep Kit (New England BioLabs) according to the manufacturer's protocol, and the libraries were individually indexed with the NEBNext Multiplex Oligos for Illumina (New England BioLabs) by PCR. The quality and quantity of each amplified library were analyzed using the Bioanalyzer 2100 (Agilent Technologies, Santa Clara, CA, USA), the High Sensitivity DNA kit (Agilent Technologies), and the NEBNext Library Quant Kit for Illumina (New England BioLabs).

### Genome size estimation

The genome size of *Q. glauca* was estimated by GenomeScope2 v2.0 (*Ranallo-Benavidez et al., 2020*) using *k*-mer frequency plot obtained by kmc v3.2.4 (*Kokot et al., 2017*) with *k*-mer size 61 against the HiFi reads. We also used paired-end short reads for genome size estimation of *Q. glauca* and confirmed that the resulting estimates were not different. For *L. edulis*, paired-end short reads from four individuals were merged and quality-trimmed using fastp v0.23.2 (*Chen et al., 2018*). The genome size was then estimated using KmerGenie (*Chikhi and Medvedev, 2014*) with a k-mer size of 55 applied to the trimmed reads.

### Phylogenetic reconstruction

We constructed a phylogenetic tree using an alignment-free method (*Fan et al., 2015*) based on short-read data from *Q. acuta* and *L. glaber* and genome assembly data from *Q. glauca* and *L. edulis*. Additionally, genomic data for *Fagus sylvatica* was obtained from https://www.beechgenome.net/ (*Mishra et al., 2021*) and included as an outgroup. Due to the large volume of short-read data for *Q. acuta* and *L. glaber* (18 Gb each), we randomly subsampled half of the reads. Bootstrap values were estimated through 100 resampling of the k-mer table.

### Identification of orthologs

The prediction of orthogroups was based on a BLASTp all-against-all comparison of the protein sequences (E-value<$10^{-5}$) of 5 Fagaceae species: *L. edulis*, *Q. glauca*, *Q. robur*, *Fagus sylvatica,* and *Juglans regia*. Clustering was conducted with OrthoFinder2, using version 2.2.6 for BLAST data generation and version 2.5.5 for orthogroup determination (*Emms and Kelly, 2019*). The genomic data of *Q. robur* and *J. regia* were downloaded from NCBI (dhQueRobu3.1 and Walnut 2.0), while *F. sylvatica* data were sourced from https://www.beechgenome.net/ (*Mishra et al., 2021*). Since the *F. sylvatica* dataset contained numerous predicted transposable elements (TEs), we re-annotated genes using three databases—EggNOG5 (EggNOG-mapper), Pfam35.0 (HMMER v3.3.2), and UniRef90_2023_0628 (DIAMOND v2.1.8.162)—and excluded genes identified as TEs in at least one database. In total, we identified 26692 orthogroups and focused on single-copy orthogroups after filtering out genes mapped outside the 12 chromosomes. This resulted in a final dataset of 11,749 genes for further analysis (Data S2).

### Hierarchical clustering and PCA

To assess the similarity of the genome-wide transcriptional profiles across orthologous genes and 483 samples collected in different seasons, we performed hierarchical clustering using the monthly time series data of 11,749 orthologous genes from October 2019 to April 2023. For each orthologous gene, there were 26 or 27 time points, with three individuals each for *Q. glauca*, *Q. acuta*, *L. glaber*, and one individual for *L. edulis*. We calculated the mean expression levels of each orthologous gene across three individuals in each species and subsequently normalized the values by adjusting the mean to zero and the standard deviation to one. We performed hierarchical clustering using the Ward method and the Euclidean distance using the pheatmap function in R (v 1.0.12). To determine the number of the clusters, we calculated Within-Cluster-Sum of Squared Error (WSS) and determined the number of clusters as an elbow point of it (*Figure 2—figure supplement 4*). To assess the seasonal expression dynamics of 11,749 orthologous genes, we also performed PCA of the gene expression profiles from all samples using the prcomp function in R (v 4.3.1).

### Identification of genes with rhythmic expression

To identify genes with rhythmic expression and determine the period and peak month of gene expression, we applied the RAIN algorithm (*Thaben and Westermark, 2014*) using the rain function in R (v 1.34.0). We set the following settings: deltat = 1, period = 12, period.delta=11, peak.border =

(0,1), nr.series=1, method = 'independent,' na.rm=T, adjp.method = 'ABH.' Based on the results of the RAIN analysis, we identified genes with significant periodicity using a Benjamini-Hochberg (BH) adjusted $q$-value threshold of <0.01. Since the period length and peak month in the RAIN output are expressed in terms of the number of time points, we converted them into months using the sampling span (four weeks) and sampling dates. Subsequently, we categorized the rhythmic genes into three categories based on the period length $T$ (in months): 'half-annual' ($0 \leq T < 8$), 'annual' ($8 \leq T \leq 16$), and 'long' ($T > 16$).

## Calculation of Pearson's correlation coefficient across species

To quantify the similarity of seasonal gene expression pattern across species, we calculated pairwise Pearson's correlations of seasonal expression for each gene across all the species pairs. For species within the same genus, the monitoring period was identical, allowing direct comparison. On the other hand, the monitoring periods differed between *Quercus* and *Lithocarpus* species: *Q. glauca* and *Q. acuta* were monitored from May 2021 to April 2023, while *L. edulis* and *L. glaber* were monitored from October 2019 to October 2021 (*Figure 2—figure supplement 3*). To ensure comparable time series data for inter-genera comparisons, we extracted the time series from *Q. glauca* and *Q. acuta* spanning May 19, 2021 to November 2, 2022 and from *L. edulis* and *L. glaber* spanning May 13, 2020 to October 20, 2021 for correlation analysis. The mean absolute differences for each gene across all the species pairs were also calculated in a similar manner.

## Molecular phenology divergence index (*D*)

To quantify the seasonal difference of peak month divergence across species, we defined the molecular phenology divergence index (*D*) for each month $t$ and species pair A-B as the proportion of genes that peak in month $t$ in the reference species A but exhibit a peak month difference ($\Delta\varphi$) greater than two months in species B using the following formula:

$$D : = \frac{\# \text{Genes with peak month } t \text{ in species A and } t \pm \Delta\phi \, (\geq 2) \text{ in species B}}{\# \text{Genes with peak month } t \text{ in species A}}.$$

## GO enrichment analysis

First, we assigned GO terms for each gene in *Lithocarpus* and *Quercus* genomes based on the annotation data of *Arabidopsis thaliana* orthologs downloaded from PLAZA Database (http://bioinformatics.psb.ugent.be/plaza/). Then, GO enrichment analysis was conducted using the GOstats (*Falcon and Gentleman, 2007*) function in R (v 2.66.0) with the following settings: ontology = BP, pvalueCutoff = 0.05, conditional = TRUE, and testDirection = over. The universe gene set was defined as genes associated with GO terms common to both of the genomes. Genes without annotated GO terms were removed from the universe set, and those with differing GO term annotations between the two genomes were also excluded. The BH-adjusted $q$-values were calculated from the $p$-values generated by GOstats to account for multiple testing.

## Phenology measurement: leaf unfolding and flowering

To compare seasonal changes in genome-wide gene expression profiles with phenology, we conducted observations of leaf unfolding and flowering phenology on the three terminal branches of each of the three individuals used for gene expression monitoring. Leaf unfolding was defined as the stage where both the entire leaf blade and the leaf stalk were visible (*Figure 2—figure supplement 5*). Flowering was defined as the stage where either male or female flowers were visible (*Figure 2—figure supplement 5*). We subsequently plotted the proportion of branches exhibiting leaf unfolding and flowering. The data are provided in *Supplementary file 3*.

## Estimation of protein-coding sequence divergence

Pairwise distances for non-synonymous (dN) and synonymous (dS) substitutions, as well as the dN/dS ratio, were estimated for each orthologous gene pair between *L. edulis* and *Q. glauca* using codeml (PAML 4.0) (*Yang, 2007*). The analysis was performed with the following settings: seqtype = 1, Codon-Freq = 2, Runmode = –2, and the transition/transversion ratio estimated from the data. To ensure reliability, genes exhibiting signs of saturated divergence or underestimated dS values were excluded,

as codeml produces robust results only within a moderate range of sequence divergence. Specifically, we removed 43 orthologs with dN/dS ≥ 99 and 483 genes with dS >10, dS <0.001, or dN <0.0003. The thresholds of dN and dS were determined by examining the relationship between dN and dS and excluding outliers that deviated from the main distribution. After filtering, the final dataset comprised 11,223 orthologous genes.

Linear regression analyses were performed to examine the relationship between dN, dS, dN/dS, and peak month differences ($\Delta\varphi$) in gene expression between *Q. glauca* and *L. edulis*, as well as mean expression levels. These analyses were conducted using the lm function in R software (v 4.3.1). Specifically, we modeled the evolutionary rates (dN, dS, and dN/dS) as the response variable and the peak month difference ($\Delta\varphi$) or the mean levels in gene expression as the predictor variable. To account for the skewed distributions of evolutionary rates, logarithm transformations with base 2 were applied.

## Acknowledgements

We thank Atsuko Miyawaki-Kuwakado for her valuable technical advice on RNA-seq analysis. We also appreciate the technical support provided by Kayoko Ohta. Additionally, we thank Koharu Yamaguchi, Yuta Aoyagi, and Ryosuke Imai for their help with sample collection. This work was supported by JSPS KAKENHI Grant Numbers JP23H04965 and JP23H04966.

## Additional information

### Funding

| Funder | Grant reference number | Author |
|---|---|---|
| Japan Society for the Promotion of Science | JP23H04965 | Akiko Satake |
| Japan Society for the Promotion of Science | JP23H04966 | Akiko Satake |

The funders had no role in study design, data collection and interpretation, or the decision to submit the work for publication.

### Author contributions

Shuichi N Kudo, Conceptualization, Data curation, Formal analysis, Investigation, Visualization, Methodology, Writing - original draft, Writing – review and editing; Yuka Ikezaki, Data curation, Formal analysis, Investigation, Visualization, Methodology, Writing – review and editing; Junko Kusumi, Hideki Hirakawa, Sachiko Isobe, Data curation, Formal analysis, Investigation, Methodology, Writing – review and editing; Akiko Satake, Conceptualization, Resources, Formal analysis, Supervision, Funding acquisition, Investigation, Methodology, Writing - original draft, Project administration, Writing – review and editing

### Author ORCIDs

Shuichi N Kudo https://orcid.org/0009-0003-9385-1794
Yuka Ikezaki https://orcid.org/0009-0003-8805-124X
Akiko Satake https://orcid.org/0000-0002-0831-8617

Reviewer #2 (Public review): https://doi.org/10.7554/eLife.107309.3.sa1
Author response https://doi.org/10.7554/eLife.107309.3.sa2

## Additional files

### Supplementary files

Supplementary file 1. Statistics for genome assemblies of two new sequenced species.

Supplementary file 2. Summary of clustering 213 gene expression profiles.

Supplementary file 3. Hierarchical clustering of 11,749 gene expression dynamics.

Supplementary file 4. Identification of genes with rhythmic expression.

Supplementary file 5. Comparison of correlation and absolute difference among the species pairs.

Supplementary file 6. List of genes with conserved and diverged gene expression across species.

Supplementary file 7. Statistical test of evolutionary rates.

MDAR checklist

## Data availability

All sequencing data generated in this study have been deposited in the DNA Data Bank of Japan (DDBJ) and are available under the BioProject accession number PRJDB20498. RNA-seq reads used for expression analyses are available under the DDBJ Sequence Read Archive (DRA) run accession numbers DRR659185-DRR659374, DRR664882-DRR665873, and DRR724385-DRR724496, while DNA sequences used for genome assembly, genome size estimation and phylogenetic reconstruction are available under DRR659899-DRR659929. The assembled genomes are also available from DDBJ: the Q. glauca genome under accession numbers BAAHOF010000001-BAAHOF010000797, and the L. edulis genome under BAAHOE010000001-BAAHOE010000750. All codes used in the analysis are available at GitHub, copy archived at *Kudo, 2025*.

The following dataset was generated:

| Author(s) | Year | Dataset title | Dataset URL | Database and Identifier |
|---|---|---|---|---|
| Kudo SN, Ikezaki Y, Kusumi J, Hirakawa H, Isobe S, Satake A | 2025 | Genome sequencing of Quercus glauca and Lithocarpus edulis | https://ddbj.nig.ac.jp/search/entry/bioproject/PRJDB20498 | DDBJ BioProject, PRJDB20498 |

The following previously published datasets were used:

| Author(s) | Year | Dataset title | Dataset URL | Database and Identifier |
|---|---|---|---|---|
| Carlson JE, Staton ME, Addo-Quaye C, Cannon N, Zhebentyayeva T, Islam-Faridi N, Yu J, Huff M, Fan S, Conrad AO, Schuster SC, Abbott AG, Westbrook J, Holliday J, Nelson CD, Georgi L, Hebard FV | 2020 | Castanea mollissima genome assembly ASM76360v2 | https://www.ncbi.nlm.nih.gov/datasets/genome/GCA_000763605.2/ | NCBI GenBank, GCA_000763605.2 |
| Sork VL, Fitz-Gibbon ST, Puiu D, Crepeau M, Gugger PF, Sherman R, Stevens K, Langley CH, Pellegrini M, Salzberg SL | 2019 | Quercus lobata genome assembly ValleyOak3.0 | https://www.ncbi.nlm.nih.gov/datasets/genome/GCF_001633185.1/ | NCBI RefSeq, GCF_001633185.1 |
| Wellcome Sanger Tree of Life Programme | 2022 | Quercus robur reference genome dhQueRobu3.1 | https://www.ncbi.nlm.nih.gov/datasets/genome/GCF_932294415.1/ | NCBI RefSeq, GCF_932294415.1 |

*Continued on next page*

*Continued*

| Author(s) | Year | Dataset title | Dataset URL | Database and Identifier |
|---|---|---|---|---|
| Ramos AM, Barros PM, Capote T, Chaves I, Simões F, Abreu I, Carrasquinho I, Faro C, Guimarães JB, Mendonça D, Nóbrega F, Rodrigues L, Saibo NJM, Varela MC, Egas C, Matos J, Miguel CM, Oliveira MM, Ricardo CP, Gonçalves S, Usié BP | 2018 | Genome assembly CorkOak1.0 | https://www.ncbi.nlm.nih.gov/datasets/genome/GCF_002906115.1/ | NCBI RefSeq, GCF_002906115.1 |
| Yang X, Wang Z, Zhang L, Hao G, Liu J, Yang Y | 2019 | Genome assembly ASM693729v1 | https://www.ncbi.nlm.nih.gov/datasets/genome/GCA_006937295.1/ | NCBI GenBank, GCA_006937295.1 |
| Lovell JT, Bentley NB, Bhattarai G, Jenkins JW, Sreedasyam A, Alarcon Y, Bock C, Boston LB, Carlson J, Cervantes K, Clermont K, Duke S, Krom N, Kubenka K, Mamidi S, Mattison CP, Monteros MJ, Pisani C, Plott C, Rajasekar S, Rhein HS, Rohla C, Song M, Hilaire RS, Shu S, Wells L, Webber J, Heerema RJ, Klein PE, Conner P, Wang X, Grauke LJ, Grimwood J, Schmutz J, Randall JJ | 2021 | Genome assembly C.illinoinensisPawnee_v1 | https://www.ncbi.nlm.nih.gov/datasets/genome/GCF_018687715.1/ | NCBI RefSeq, GCF_018687715.1 |
| Zhu T, Wang L, You FM, Rodriguez JC, Deal KR, Chen L, Li J, Chakraborty S, Balan B, Jiang CZ, Brown PJ, Leslie CA, Aradhya MK, Dandekar AM, McGuire PE, Kluepfel D, Dvorak J, Luo MC | 2019 | Genome assembly Jm3101_v1.0 | https://www.ncbi.nlm.nih.gov/datasets/genome/GCF_004785595.1/ | NCBI RefSeq, GCF_004785595.1 |
| Peng S, Yang G, Liu C, Yu Z, Zhai M | 2020 | Genome assembly Walnut 2.0 | https://www.ncbi.nlm.nih.gov/datasets/genome/GCF_001411555.2/ | NCBI RefSeq, GCF_001411555.2 |
| Sork VL, Fitz-Gibbon ST, Puiu D, Crepeau M, Gugger PF, Sherman R, Stevens K, Langley CH, Pellegrini M, Salzberg SL | 2019 | Genome assembly ValleyOak3.2 | https://www.ncbi.nlm.nih.gov/datasets/genome/GCF_001633185.2/ | NCBI RefSeq, GCF_001633185.2 |
| Satake A, Ohta K, Takeda-Kamiya N, Toyooka K, Kusumi J | 2024 | Quercus glauca transcriptome project | https://www.ncbi.xyz/geo/query/acc.cgi?acc=GSM6468274 | NCBI Gene Expression Omnibus, GSE211384 |

*Continued on next page*

*Continued*

| Author(s) | Year | Dataset title | Dataset URL | Database and Identifier |
|---|---|---|---|---|
| Jia HM, Jia HJ, Cai QL, Wang Y, Zhao HB, Yang WF, Wang GY, Zhan DL, Shen YT, Niu QF, Chang L, Qiu J, Zhao L, Xie HB, Jin J, Jiao Y, Zhou CC, Tu T, Chai CY, Gao JL, Fan LJ, Weg E, Wang JY, Gao ZS, Li YH, Li XW, Fu WY | 2020 | Genome assembly Mru_ZJU_2 | https://www.ncbi.nlm.nih.gov/datasets/genome/GCA_003952965.2/ | NCBI GenBank, GCA_003952965.2 |

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
