## [Editor Report · eLife Assessment]

The authors collected time-course RNA-seq data from four tree species in natural environments and analyzed seasonal patterns of gene expression. This **fundamental** study substantially advances our understanding of how seasonal environments shape gene expression. The evolutionary effects of seasonal environments on gene expression are rarely studied at this scale and the dataset is extensive. The evidence supporting the conclusions is **compelling**, with caveats and limitations clearly described. The work will be of broad interest to colleagues studying evolution and gene expression.

---

## [Referee Report · Reviewer #2 (Public review)]

This study investigates how seasonal environments shape the evolution of gene expression by analyzing two-year time-series transcriptomes from leaves and buds of four Fagaceae tree species. The revised manuscript incorporates additional data and analyses that directly address earlier concerns about sampling design and environmental variation, thereby strengthening the robustness of the conclusions.

The major strengths of this work are the scale and quality of the dataset, the integration of genome assemblies with time-series transcriptomics, and the careful analyses showing that winter bud expression is strongly conserved across species. The additional samples and re-analyses demonstrate convincingly that these results are not artifacts of sampling period or site differences. The study also links gene expression dynamics to phenological observations and frames its findings in relation to broader evolutionary concepts such as phenological synchrony and the developmental hourglass model.

Remaining limitations include the absence of direct mechanistic analyses of cis-regulatory and chromatin-level processes, the relatively coarse resolution of phenological trait measurements, and the weak association between seasonal expression divergence and sequence divergence. Importantly, these limitations are now explicitly acknowledged in the revised Discussion and framed as directions for future research.

Overall, the authors have substantially achieved their aims. This revised version represents a robust and convincing contribution that provides valuable data resources and conceptual insights into how seasonal environments constrain and shape gene expression. It will be of interest not only to evolutionary biologists and plant scientists, but also to researchers considering the broader role of environmental cycles in gene regulatory evolution.

---

## [Author Response]

The following is the authors’ response to the original reviews

**Public Reviews**

**Reviewer #1 (Public review):**
Summary:The authors performed genome assemblies for two Fagaceae species and collected transcriptome data from four natural tree species every month over two years. They identified seasonal gene expression patterns and further analyzed species-specific differences.Strengths:The study of gene expression patterns in natural environments, as opposed to controlled chambers, is gaining increasing attention. The authors collected RNA-seq data monthly for two years from four tree species and analyzed seasonal expression patterns. The data are novel. The authors could revise the manuscript to emphasize seasonal expression patterns in three species (with one additional species having more limited data). Furthermore, the chromosome-scale genome assemblies for the two Fagaceae species represent valuable resources, although the authors did not cite existing assemblies from closely related species.

Thank you for your careful assessment of our manuscript.

Weaknesses:Comment; The study design has a fundamental flaw regarding the evaluation of genetic or evolutionary effects. As a basic principle in biology, phenotypes, including gene expression levels, are influenced by genetics, environmental factors, and their interaction. This principle is well-established in quantitative genetics.In this study, the four species were sampled from three different sites (see Materials and Methods, lines 543-546), and additionally, two species were sampled from 2019-2021, while the other two were sampled from 2021-2023 (see Figure S2). This critical detail should be clearly described in the Results and Materials and Methods. Due to these variations in sampling sites and periods, environmental conditions are not uniform across species.Even in studies conducted in natural environments, there are ways to design experiments that allow genetic effects to be evaluated. For example, by studying co-occurring species, or through transplant experiments, or in common gardens. To illustrate the issue, imagine an experiment where clones of a single species were sampled from three sites and two time periods, similar to the current design. RNA-seq analysis would likely detect differences that could qualitatively resemble those reported in this manuscript.One example is in line 197, where genus-specific expression patterns are mentioned. While it may be true that the authors' conclusions (e.g., winter synchronization, phylogenetic constraints) reflect real biological trends, these conclusions are also predictable even without empirical data, and the current dataset does not provide quantitative support.If the authors can present a valid method to disentangle genetic and environmental effects from their dataset, that would significantly strengthen the manuscript. However, I do not believe the current study design is suitable for this purpose.Unless these issues are addressed, the use of the term "evolution" is inappropriate in this context. The title should be revised, and the result sections starting from "Peak months distribution..." should be either removed or fundamentally revised. The entire Discussion section, which is based on evolutionary interpretation, should be deleted in its current form.If the authors still wish to explore genetic or evolutionary analyses, the pair of L. edulis and L. glaber, which were sampled at the same site and over the same period, might be used to analyze "seasonal gene expression divergence in relation to sequence divergence." Nevertheless, the manuscript would benefit from focusing on seasonal expression patterns without framing the study in evolutionary terms.

We sincerely thank the reviewer for the detailed and thoughtful comments. We fully recognize the importance of carefully distinguishing genetic and environmental contributions in transcriptomic studies, particularly when addressing evolutionary questions. The reviewer identified two major concerns regarding our study design: (1) the use of different monitoring periods across species, and (2) the use of samples collected from different study sites. We addressed both concerns with additional analyses using 112 new samples and now present new evidence that supports the robustness of our conclusions.

(1) Monitoring period variation does not bias our conclusions

To address concerns about the differing monitoring periods, we added new RNA-seq data (42 samples each for bud and leaf samples for *L. glaber* and 14 samples each for bud and leaf samples for _L. eduli_s) collected from November 2021 to November 2022, enabling direct comparison across species within a consistent timeframe. Hierarchical clustering of this expanded dataset (Fig. S6) yielded results consistent with our original findings: winter-collected samples cluster together regardless of species identity. This strongly supports our conclusion that the seasonal synchrony observed in winter is not an artifact of the monitoring period and demonstrates the robustness of our conclusions across datasets.

(2) Site variation is limited and does not confound our findings*Q. acuta* introduced confounding environmental effects, we reanalyzed the data after excluding this species. Hierarchical clustering still revealed that winter bud samples formed a distinct cluster regardless of species identity (Fig. S7), consistent with our original finding.

Although the study included three sites, two of them (Imajuku and Ito Campus) are only 7.3 km apart, share nearly identical temperature profiles (see Fig. S2), and are located at the edge of similar evergreen broadleaf forests. Only *Q. acuta* was sampled from a higher-altitude, cooler site. To assess whether the higher elevation site of

Furthermore, we recalculated the molecular phenology divergence index *D* (Fig. 4C) and the interspecific Pearson’s correlation coefficients (Fig. 5A) without including *Q. acuta*. These analyses produced results that were similar to those obtained from the full dataset (Fig. S12; Fig. S14), indicating that the observed patterns are not driven by environmental differences associated with elevation.

(3) Justification for our approach in natural systems https://doi.org/10.1111/pce.14716 and clonal https://doi.org/10.1002/ppp3.10548 to examine environmental effects on gene expression. However, extending these approaches to long-lived tree species in diverse natural ecosystems poses significant logistical and biological challenges. In this study, we addressed this limitation by including three co-occurring species at the same site, which allowed us to evaluate interspecific differences under comparable environmental conditions. Importantly, even when we limited our analyses to these co-occurring species, the results remained consistent, indicating that the observed variation in transcriptomic profiles cannot be attributed to environmental factors alone and likely reflects underlying genetic influences.

We agree with the reviewer that experimental approaches such as common gardens, reciprocal transplants, and the use of co-occurring species are valuable for disentangling genetic and environmental effects. In fact, we have previously implemented such designs in studies using the perennial herb *Arabidopsis halleri* (Komoto et al., 2022), *Someiyoshino* cherry trees Miyawaki-Kuwakado et al., 2024,

Accordingly, we added four new figures (Fig. S6, Fig. S7, Fig. S12 and Fig. S14) and revised the manuscript to clarify the limitations and strengths of our design, to tone down the evolutionary claims where appropriate, and to more explicitly define the scope of our conclusions in light of the data. We hope that these efforts sufficiently address the reviewer’s concerns and strengthen the manuscript.

To better support the seasonal expression analysis, the early RNA-seq analysis sections should be strengthened. There is little discussion of biological replicate variation or variation among branches of the same individual. These could be important factors to analyze. In line 137, the mapping rate for two species is mentioned, but the rates for each species should be clearly reported. One RNA-seq dataset is based on a species different from the reference genome, so a lower mapping rate is expected. While this likely does not hinder downstream analysis, quantification is important.

We thank the reviewer 1 for the helpful comment. To evaluate the variation among biological replicates, we compared the expression level of each gene across different individuals. We observed high correlation between each pair of individuals *Q. glauca* (*n=3*): an average correlation coefficient *r* = 0.947; *Q. acuta* (*n=3*): *r* = 0.948; *L. glaber* (*n=3*): *r* = 0.948. This result suggests that the seasonal gene expression pattern is highly synchronized across individuals within the same species. We mentioned this point in the Result section in the revised manuscript. We also calculated the mean mapping rates for each species. As the reviewer expected, the mapping rate was slightly lower in *Q. acuta* (88.6 ± 2.3%) and *L. glaber* (84.3 ± 5.4%), whose RNA-Seq data were mapped to reference genomes of related but different species, compared to that in *Q. glauca* (92.6 ± 2.2%) and *L. edulis* (89.3 ± 2.7%). However, we minimized the impact of these differences on downstream analysis. These details have been included in the revised main text.

In Figures 2A and 2B, clustering is used to support several points discussed in the Results section (e.g., lines 175-177). However, clustering is primarily a visualization method or a hypothesis-generating tool; it cannot serve as a statistical test. Stronger conclusions would require further statistical testing.

We thank the reviewer for the helpful comment. As noted, we acknowledge that hierarchical clustering (Fig. 2A) is primarily a visualization and hypothesis-generating method. To assess the biological relevance of the clusters identified, we conducted a Mann-Whitney U test or the Steel-Dwass test to evaluate whether the environmental temperatures at the time of sample collection differed significantly among the clusters. This analysis (Fig. 2B) revealed statistically significant differences in temperature in the cluster B3 (*p* < 0.01), indicating that the gene expression clusters are associated with seasonal thermal variation. These results support the interpretation that the clusters reflect coordinated transcriptional responses to environmental temperature. We revised the Results section to clarify this point.

The quality of the genome assemblies appears adequate, but related assemblies should be cited and discussed. Several assemblies of Fagaceae species already exist, including Quercus mongolica (Ai et al., Mol Ecol Res, 2022), Q. gilva (Front Plant Sci, 2022), and Fagus sylvatica (GigaScience, 2018), among others. Is there any novelty here? Can you compare your results with these existing assemblies?

We agree that genome assemblies of Fagaceae species are becoming increasing available. However, our study does not aim to emphasize the novelty of the genome assemblies per se. Rather, with the increasing availability of chromosome-level genomes, we regard genome assembly as a necessary foundation for more advanced analyses. The main objective of our study is to investigate how each gene is expressed in response to seasonal environmental changes, and to link genome information with seasonal transcriptomic dynamics. To address the reviewer’s comment in line with this objective, we added a discussion on the syntenic structure of eight genome assemblies spanning four genera within the Fagaceae, including a species from the genus *Fagus* (Ikezaki et al. 2025, https://doi.org/10.1101/2025.07.31.667835). This addition helps to position our work more clearly within the context of existing genomic resources.

Most importantly, Figure 1B-D shows synteny between the two genera but also indicates homology between different chromosomes. Does this suggest paleopolyploidy or another novel feature? These chromosome connections should be interpreted in the main text-even if they could be methodological artifacts.

A previous study on genome size variation in Fagaceae suggested that, given the consistent ploidy level across the family, genome expansion likely occurred through relatively small segmental duplications rather than whole-genome duplications. Because Figure 1B-D supports this view, we cited the following reference in the revised version of the manuscript. Chen et al. (2014) https://doi.org/10.1007/s11295-014-0736-y

In both the Results and Materials and Methods sections, descriptions of genome and RNA-seq data are unclear. In line 128, a paragraph on genome assembly suddenly introduces expression levels. RNA-seq data should be described before this. Similarly, in line 238, the sentence "we assembled high-quality reference genomes" seems disconnected from the surrounding discussion of expression studies. In line 632, Illumina short-read DNA sequencing is mentioned, but it's unclear how these data were used.

We relocated the explanation regarding the expression levels of single-copy and multi-copy genes to the section titled “Seasonal gene expression dynamics.” Additionally, we clarified in the Materials and Methods section that short-read sequencing data were used for both genome size estimation and phylogenetic reconstruction.

**Reviewer #2 (Public review):**
Summary:This study explores how gene expression evolves in response to seasonal environments, using four evergreen Fagaceae species growing in similar habitats in Japan. By combining chromosome-scale genome assemblies with a two-year RNA-seq time series in leaves and buds, the authors identify seasonal rhythms in gene expression and examine both conserved and divergent patterns. A central result is that winter bud expression is highly conserved across species, likely due to shared physiological demands under cold conditions. One of the intriguing implications of this study is that seasonal cycles might play a role similar to ontogenetic stages in animals. The authors touch on this by comparing their findings to the developmental hourglass model, and indeed, the recurrence of phenological states such as winter dormancy may act as a cyclic form of developmental canalization, shaping expression evolution in a way analogous to embryogenesis in animals.Strengths:(1) The evolutionary effects of seasonal environments on gene expression are rarely studied at this scale. This paper fills that gap.(2) The dataset is extensive, covering two years, two tissues, and four tree species, and is well suited to the questions being asked.(3) Transcriptome clustering across species (Figure 2) shows strong grouping by season and tissue rather than species, suggesting that the authors effectively controlled for technical confounders such as batch effects and mapping bias.(4) The idea that winter imposes a shared constraint on gene expression, especially in buds, is well argued and supported by the data.(5) The discussion links the findings to known concepts like phenological synchrony and the developmental hourglass model, which helps frame the results.

We are grateful for the reviewer for the detailed and thoughtful review of our manuscript.

Weaknesses:(1) While the hierarchical clustering shown in Figure 2A largely supports separation by tissue type and season, one issue worth noting is that some leaf samples appear to cluster closely with bud samples. The authors do not comment on this pattern, which raises questions about possible biological overlap between tissues during certain seasonal transitions or technical artifacts such as sample contamination. Clarifying this point would improve confidence in the interpretation of tissue-specific seasonal expression patterns.

Leaf samples clustered into the bud are newly flushed leaves collected in April for *Q. glauca*, May for *Q. acuta*, May and June for *L. edulis*, and August and September for *L. glaber*. To clarify this point, we highlighted these newly flushed leaf samples as asterisk in the revised figure (Fig. 2A).

(2) While the study provides compelling evidence of conserved and divergent seasonal gene expression, it does not directly examine the role of cis-regulatory elements or chromatin-level regulatory architecture. Including regulatory genomic or epigenomic data would considerably strengthen the mechanistic understanding of expression divergence.

We thank the reviewer for this insightful comment. As noted in the Discussion section, we hypothesize that such genome-wide seasonal expression patterns—and their divergence across species—are likely mediated by cis-regulatory elements and chromatin-level mechanisms. While a direct investigation of regulatory architecture was beyond the scope of the present study, we fully agree that incorporating regulatory genomic and epigenomic data would significantly deepen the mechanistic understanding of expression divergence. In this regard, we are currently working to identify putative cis-regulatory elements in non-coding regions and are collecting epigenetic data from the same tree species using ChIP-seq. We believe the current study provide a foundation for these future investigations into the regulatory basis of seasonal transcriptome variation. We made a minor revision to the Discussion to note that an important future direction is to investigate the evolution of non-coding sequences that regulate gene expression in response to seasonal environmental changes.

(3) The manuscript includes a thoughtful analysis of flowering-related genes and seasonal GO enrichment (e.g., Figure 3C-D), providing an initial link between gene expression timing and phenological functions. However, the analysis remains largely gene-centric, and the study does not incorporate direct measurements of phenological traits (e.g., flowering or bud break dates). As a result, the connection between molecular divergence and phenotypic variation, while suggestive, remains indirect.

We would like to note that phenological traits have been observed in the field on a monthly basis throughout the sampling period and the phenological data were plotted together with molecular phenology (e.g. Fig. 2A, C; Fig. 3C, D). Although the temporal resolution is limited, these observations captured species-specific differences in key phenological events such as leaf flushing and flowering times. We revised the manuscript to clarify this point.

(4) Although species were sampled from similar habitats, one species (Q. acuta) was collected at a higher elevation, and factors such as microclimate or local photoperiod conditions could influence expression patterns. These potential confounding variables are not fully accounted for, and their effects should be more thoroughly discussed or controlled in future analyses.

We fully agree with the reviewer that local environmental conditions, including microclimate and photoperiod differences, could potentially influence gene expression patterns. To assess whether the higher elevation site of *Q. acuta* introduced confounding environmental effects, we reanalyzed the data after excluding this species. Hierarchical clustering still revealed that winter bud samples formed a distinct cluster regardless of species identity (Fig. S7), consistent with our original finding.

Furthermore, we recalculated the molecular phenology divergence index D (Fig. 4C) and the interspecific Pearson’s correlation coefficients (Fig. 5A) without including *Q. acuta*. These analyses produced results that were qualitatively similar to those obtained from the full dataset (Fig. S12; Fig. S14), indicating that the observed patterns are not driven by environmental differences associated with elevation.

We believe these additional analyses help to decouple the effects of environment and genetics, and support our conclusion that both seasonal synchrony and phylogenetic constraints play key roles in shaping transcriptome dynamics. We added four new figures (Fig. S6, Fig. S7, Fig. S12 and Fig. S14) and revised the text accordingly to clarify this point and to acknowledge the potential impact of site-specific environmental variation.

(5) Statistical and Interpretive Concerns Regarding Δφ and dN/dS Correlation (Figures 5E and 5F):a) Statistical Inappropriateness: Δφ is a discrete ordinal variable (likely 1-11), making it unsuitable for Pearson correlation, which assumes continuous, normally distributed variables. This undermines the statistical validity of the analysis.

We thank the reviewer for the insightful comment. We would like to clarify that the analysis presented in Figures 5E and 5F was based on linear regression, not Pearson’s correlation. Although Δ*φ* is a discrete variable, it takes values from 0 to 6 in 0.5 increments, resulting in 13 levels. We treated it as a quasi-continuous variable for the purposes of linear regression analysis. This approach is commonly adopted in practice when a discrete variable has sufficient resolution and ordering to approximate continuity. To enhance clarity, we revised the manuscript to explicitly state that linear regression was used, and we now reported the regression coefficient and associated *p*-value to support the interpretation of the observed trend.

b) Biological Interpretability: Even with the substantial statistical power afforded by genome-wide analysis, the observed correlations are extremely weak. This suggests that the relationship, if any, between temporal divergence in expression and protein-coding evolution is negligible.Taken together, these issues weaken the case for any biologically meaningful association between Δφ and dN/dS. I recommend either omitting these panels or clearly reframing them as exploratory and statistically limited observations.

We agree with the reviewer’s comment. While we retained the original panels, we reframed our interpretation to emphasize that, despite statistical significance, the observed correlation is very weak—suggesting that coding region variation is unlikely to be the primary driver of seasonal gene expression patterns. Accordingly, we revised the “Relating seasonal gene expression divergence to sequence divergence” section in the Results, as well as the relevant part of the Discussion.

**Recommendations for the authors:**

**Reviewer #1 (Recommendations for the authors):**
Sentences around lines 250-251 are incomplete and need revision.

We thank the reviewer for pointing this out. We revised the sentences in the subsection “Peak month distribution of rhythmic genes and intra-genus and inter-genera comparison” in the Results section to ensure clarity and completeness. In addition, to improve the interpretability of the peak month distribution, we added arrows to indicate the major peaks in the circular histograms shown in Fig. 3C and 3D.

**Reviewer #2 (Recommendations for the authors):**
(1) In Figure 1E-G, the term Copy number or Copy number variation could be misleading, as it is commonly associated with inter-individual gene copy number variation in a population. Since the analysis here refers to orthology relationships rather than population-level variation, a more precise term, such as orthogroup classification, may be preferable.

We thank the reviewer for this helpful suggestion. We agree that the term “copy number” could be misleading in this context. Accordingly, we updated the labeling in Fig. 1 to reflect the more precise term “orthogroup classification.”

(2) In Figure 3A, the x-axis label Period (month) may be misleading, as it could be mistaken for calendar months rather than referring to the periodicity of gene expression cycles. A more explicit label, such as Expression periodicity (months), might improve clarity for the reader.

We thank the reviewer for this valuable suggestion. In the original version of Fig. 3A, we used the label “Period (month),” which could indeed be misinterpreted as referring to calendar months. To clarify that this axis represents the length of gene expression cycles, we revised the label to “Period length (months).” This change also aligns with the terminology used throughout the manuscript, where “Period” refers specifically to cycle length, and “Periodicity” denotes the presence or absence of rhythmic expression.

Other minor revisions

We also made minor revisions for the reference list and the grant number details, and included the accession numbers for all DNA and RNA sequence data deposited in the DNA Data Bank of Japan (DDBJ) in the Data deposition and code availability section, in addition to the BioProject ID.